# New Perspectives on the Polyak Stepsize: Surrogate Functions and Negative Results

**Francesco Orabona**
King Abdullah University of Science and Technology (KAUST)
Thuwal, 23955-6900, Kingdom of Saudi Arabia
francesco@orabona.com

**Ryan D'Orazio**
Mila Québec AI Institute, Université de Montréal
Montréal, QC, Canada
ryan.dorazio@mila.quebec

## Abstract

The Polyak stepsize has been proven to be a fundamental stepsize in convex optimization, giving near optimal gradient descent rates across a wide range of assumptions. The universality of the Polyak stepsize has also inspired many stochastic variants, with theoretical guarantees and strong empirical performance. Despite the many theoretical results, our understanding of the convergence properties and shortcomings of the Polyak stepsize or its variants is both incomplete and fractured across different analyses. We propose a new, unified, and simple perspective for the Polyak stepsize and its variants as gradient descent on a surrogate loss. We show that *each variant is equivalent to minimize a surrogate function with stepsizes that adapt to a guaranteed local curvature*. Our general surrogate loss perspective is then used to provide a unified analysis of existing variants across different assumptions. Moreover, we show a number of negative results proving that the non-convergence results in some of the upper bounds is indeed real.

## 1 Introduction

The iterative optimization of complex functions forms a cornerstone of modern machine learning, scientific computing, and engineering. Among the most foundational first-order methods is gradient descent, which iteratively refines a solution by moving in the direction opposite to the function's gradient. A critical aspect of gradient descent is the selection of an appropriate stepsize (or learning rate), as it dictates both the speed of convergence and the stability of the algorithm. The wrong choice of the stepsizes can lead to slow convergence or, conversely, to divergence, making the tuning process a significant practical hurdle.

In this landscape of stepsize selection strategies, the Polyak stepsize, proposed by Polyak [36] stands out for its theoretical elegance and convergence properties. Starting from an arbitrary $\boldsymbol{x}_1 \in \mathbb{R}^d$, the Polyak stepsize is defined as

$$\boldsymbol{x}_{t+1} = \boldsymbol{x}_t - \frac{f(\boldsymbol{x}_t) - f^\star}{\|\boldsymbol{g}_t\|^2} \boldsymbol{g}_t, \tag{1}$$

where $\boldsymbol{0} \neq \boldsymbol{g}_t \in \partial f(\boldsymbol{x}_t)$ and $f^\star = \min_{\boldsymbol{x}} f(\boldsymbol{x})$. If $\boldsymbol{g}_t = \boldsymbol{0}$, then $\boldsymbol{x}_{t+1} = \boldsymbol{x}_t$. This update rule can achieve linear convergence for strongly convex and smooth functions, $\mathcal{O}(1/T)$ rate for convex smooth functions, and $\mathcal{O}(1/\sqrt{T})$ rate for non-smooth convex ones. This is particularly interesting because all of these rates are achieved with a unique stepsize and without knowledge of smoothness

39th Conference on Neural Information Processing Systems (NeurIPS 2025).

or curvature constants. In other words, this update rule is *adaptive* to the geometry of the functions to optimize.

Recently, the Polyak stepsize has seen a resurgence in the machine learning literature, with a plethora of variants. However, despite the big number of papers on this topic, one essential research question seems still to be missing: *What makes the Polyak stepsize adaptive and when can it fail?*

**Contributions.** This paper aims to provide a novel framework to understand the Polyak stepsize, providing a clear geometric explanation of its adaptivity. In particular, we show that the adaptivity is due to a simple but powerful observation: *The Polyak stepsize minimizes a surrogate objective function that is always locally smooth.* As for standard smooth functions, we will show that the knowledge of the local smoothness constant is enough to obtain the correct rates. In addition, the local smoothness will depend only on the gradient itself, removing the need to estimate it. Furthermore, we show that minimal curvature of the surrogate is inherited from the original function as well. We also use this framework to extend its core idea to a family of algorithms. Then, we will show a number of negative results when $f(\boldsymbol{x}^\star)$ is not known and for its use in the stochastic case. These negative results complete our understanding by showing that some non-vanishing terms in existing upper bounds are necessary.

## 2 Related Work

In the pionnering work of Ermol'ev [14] stepsizes of the form $\eta_t \propto {}^1/{\|\boldsymbol{g}_t\|^2}$ were proposed for non-smooth optimization. Despite the many convergence guarantees enabled by Ermol'ev [14]'s framework, in Polyak [36] it is noted that linear convergence is not possible with such stepsizes. As an alternative, Polyak suggests the stepsize (1), which is shown to converge at favourable rates with non-smooth convex functions, and strongly convex and smooth functions. In fact, contrary to common belief, Polyak [36] was the first to show linear convergence with a rate comparable to gradient descent in the smooth and strongly convex case. Furthermore, the case where $f^\star$ is estimated was also studied, showing convergence to a level set if $f^\star$ is overestimated, and best-iterate convergence to a neighbourhood if it is underestimated. The Polyak stepsize has since been extended and studied across several applications and domains.

**Non-smooth convex.** In non-smooth convex optimization, several schemes have been developed to estimate $f^\star$ on the fly [7, 6, 16, 41, 31, 27]. In the finite-sum case with interpolation, (1) and variants have been studied as an incremental subgradient method [31, 27].

**Non-expansive operators.** In the context of non-expansive operators, the update (1) has also been studied as a special case of the subgradient projector [2, 8, 10]; where it can be shown that subgradient descent with $\eta_t = {}^{(f(x)-c)_+}/{\|\boldsymbol{g}_t\|^2}$ is a quasi-firmly non-expansive operator[1] if $f$ is convex and $c \geq f^\star$, and $\boldsymbol{x} \to \boldsymbol{x}^\star$ if $f$ is continuous [2]. Moreover, in the finite-sum setting, interpolation can also be viewed as iterating different quasi-firmly non-expansive operators with a common fixed point. For example, applying a subgradient projector sequentially (i.e., cycling through the different component functions) in a way such that each function eventually gets visited guarantees that $\boldsymbol{x}_t \to \{f(\boldsymbol{y}) \leq c\}$ [8, Example 5.9.7]. For a survey on this topic see Censor [9].

**Deterministic.** In modern optimization, (1) has been shown to achieve similar rates to gradient descent in various common assumptions (e.g., Lipschitz, smoothness and strong convexity) [22], and more recently with other assumptions such as weakly convex functions [12], $(L_0, L_1)$-smooth functions [42, 17], and directional smoothness [30].

**Stochastic.** The Polyak stepsize has also been extended to the stochastic case with emphasis on applications to machine learning [39, 5, 29, 38]. The ALI-G method [5] and $\text{SPS}_{\max}$ [29] use stochastic estimates via the sampled function $f(\boldsymbol{x}, \boldsymbol{\xi})$ and its gradient $\nabla f(\boldsymbol{x}, \boldsymbol{\xi})$ to perform a Polyak-like stepsize. $\text{SPS}_{\max}$ in addition uses $\inf_x f(x, \boldsymbol{\xi})$ as an estimate to $f^\star$, and is shown to converge at fast rates without a neighbourhood under interpolation. Folowing $\text{SPS}_{\max}$ many variants have been proposed for SGD: StoPS [24], DECSPS and $\text{SPS}_{\max}^\ell$ [35], $\text{SPS}_+$ [15]. Beyond SGD, other extensions include: mirror descent [13], with preconditioning [1], with line-search [25], and with momentum [43, 40, 33].

---

[1] An operator $T$ is quasi-firmly non-expansive if for all fixed points $\boldsymbol{x}^\star$, $\|T(\boldsymbol{x}) - \boldsymbol{x}^\star\|^2 \leq \|\boldsymbol{x} - \boldsymbol{x}^\star\|^2 - \|T(\boldsymbol{x}) - \boldsymbol{x}\|^2$. Note that a quasi-firmly non-expansive operators are also referred to as *cutters*.

**Neighbourhood of Convergence.** For $\text{SPS}_{\max}$, Loizou et al. [29] proved that the suboptimality gap is only guaranteed to shrink up to a factor that depends on the loss themselves. As we will explain in Section 5.1, this is equivalent to the guarantees in online learning where the regret is proportional to the cumulative loss of the competitor, typically denoted by $L^\star$. Hence, these kind of guarantees are usually called in online learning $L^\star$ bounds [see, e.g., 34, Section 4.2].

**Surrogates.** Gower et al. [19] show that the Polyak stepsize is equivalent to online gradient descent with fixed stepsize on a sequence of adversarially chosen self-bounded surrogate losses. Differently from our framework, the adversarial nature of their losses does not allow to show that the algorithm is minimizing a fixed function. In comparison, our surrogate approach considers a fixed surrogate loss with local smoothness, where the Polyak stepsize is chosen to be the inverse of the local smoothness.

Surprisingly enough, despite the adaptivity of the Polyak stepsize across various assumptions without modification, in previous literature there is no clear explanation why this is the case.

## 3 Definitions and Notation

We will use the following notation and definitions. All the norms in this paper are L2 norms and will be denoted by $\| \cdot \|$. For a function $f : \mathbb{R}^d \to \mathbb{R}$, we define a *subgradient* of $f$ in $\boldsymbol{x} \in \mathbb{R}^d$ as a vector $\boldsymbol{g} \in \mathbb{R}^d$ that satisfies $f(\boldsymbol{y}) \geq f(\boldsymbol{x}) + \langle \boldsymbol{g}, \boldsymbol{y} - \boldsymbol{x} \rangle, \ \forall \boldsymbol{y} \in \mathbb{R}^d$. We denote the set of subgradients of $f$ in $\boldsymbol{x}$ by $\partial f(\boldsymbol{x})$. For a differentiable function we have that $\partial f(\boldsymbol{x}) = \{\nabla f(\boldsymbol{x})\}$. A function $f : V \to \mathbb{R}$, differentiable in an open set containing $V$, is *L-smooth* w.r.t. $\| \cdot \|$ if $f(\boldsymbol{y}) \leq f(\boldsymbol{x}) + \langle \nabla f(\boldsymbol{x}), \boldsymbol{y} - \boldsymbol{x} \rangle + \frac{L}{2}\|\boldsymbol{x} - \boldsymbol{y}\|^2$ for all $\boldsymbol{x}, \boldsymbol{y} \in V$.

**Definition 1.** *We say that a function $f$ has a s-sharp minimum in $\boldsymbol{x}^\star$ if*
$$f(\boldsymbol{x}) - f(\boldsymbol{x}^\star) \geq s\|\boldsymbol{x} - \boldsymbol{x}^\star\| \ .$$

Note that if $f$ has a sharp minimum then it is not differentiable at $\boldsymbol{x}^*$ [37] and if the function is also convex and $G$-Lipschitz we immediately have $G \geq s$.

**Definition 2.** *We say that a function $f : \mathbb{R}^d \to \mathbb{R}$ is L-self-bounded if*
$$\|\boldsymbol{g}\|^2 \leq 2L(f(\boldsymbol{x}) - \inf_{\boldsymbol{x}} \ f(\boldsymbol{x})), \ \forall \boldsymbol{g} \in \partial f(\boldsymbol{x}) \ .$$

It is known that $L$-smooth functions are $L$-self-bounded [see, e.g., Lemma 4 in 28], but this definition is strictly weaker because it does not assume differentiability.

## 4 Polyak Stepsize is Gradient Descent on a Surrogate Function

Let $f$ be convex and $\boldsymbol{x}^\star \in \operatorname{argmin}_{\boldsymbol{x}} \ f(\boldsymbol{x})$. Consider the following function:
$$\phi(\boldsymbol{x}) = \frac{1}{2}\left(f(\boldsymbol{x}) - f(\boldsymbol{x}^\star)\right)^2 \ . \tag{2}$$

Instead of viewing the Polyak stepsize (1) with respect to $f$ we propose to view it equivalently as a subgradient method with respect to $\phi$. By the chain rule of subgradients [2][Corollary 16.72], subgradient descent with the Polyak stepsize (1) is equivalent to subgradient descent on $\phi$ with stepsize $\eta_t = \frac{1}{\|\boldsymbol{g}_t\|^2}$. This perspective may seem superfluous, howerever, we will show that $\frac{1}{\|\boldsymbol{g}_t\|^2}$ is strongly related to a certain notion of local curvature of $\phi$, local star upper curvature.

**Definition 3** (Local star upper curvature (LSUC)). *We say that a function $f$ with minimizer $\boldsymbol{x}^\star$ has $\lambda_{\boldsymbol{y}}$-local star upper curvature (LSUC) around $\boldsymbol{y}$ if there exists $\lambda_{\boldsymbol{y}} > 0$ such that*
$$f(\boldsymbol{x}^\star) - f(\boldsymbol{y}) - \langle \boldsymbol{g}_{\boldsymbol{y}}, \boldsymbol{x}^\star - \boldsymbol{y} \rangle \geq \frac{1}{2\lambda_{\boldsymbol{y}}}\|\boldsymbol{g}_{\boldsymbol{y}}\|_2^2, \ \forall \boldsymbol{g}_{\boldsymbol{y}} \in \partial f(\boldsymbol{y}) \ .$$

Note that if the function is LSUC everywhere, then it must be convex since we assume the existence of a subgradient.[2] It is also immediate to show that convex $L$-smooth functions are also $L$-LSUC. Indeed, for convex $L$-smooth functions we have that [32, Theorem 2.1.5]
$$f(\boldsymbol{x}) - f(\boldsymbol{y}) - \langle \nabla f(\boldsymbol{y}), \boldsymbol{x} - \boldsymbol{y} \rangle \geq \frac{1}{2L}\|\nabla f(\boldsymbol{x}) - \nabla f(\boldsymbol{y})\|^2, \ \forall \boldsymbol{x}, \boldsymbol{y} \in \mathbb{R}^d \ .$$

---

[2]If $\boldsymbol{g}_t$ is not a subgradient but a directional derivative then $f$ would be guaranteed to be star-convex [26].

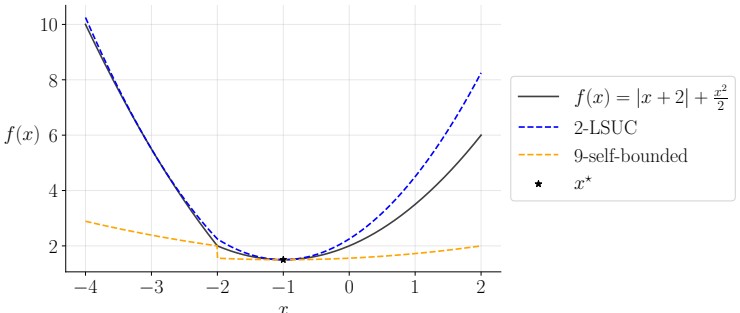

Figure 1: The function $f(x) = |x+2| + \frac{x^2}{2}$ is non-smooth but is 2-LSUC as demonstrated by the blue curve, $f(x^\star) - \langle g, x^\star - x \rangle - 1/4\|g\|^2$, being larger than $f(x)$ for all $x$ and $g \in \partial f(x)$. Similarly, $f$ is self-bounded but with the larger constant $L = 9$.

So, it is enough to set $x = x^\star$ to obtain the above definition. However, the inclusion is strict, because there exist functions that are not smooth and still satisfy the above definition. For example, as shown in Figure 1, one can easily verify that $f(x) = |x+2| + \frac{x^2}{2}$ is 2-LSUC and 9-self-bounded but not differentiable $x = -2$, hence it is not smooth.

Finally, if the star-upper-curvature holds globally, i.e., there exists $0 < \lambda < \lambda_y$ for all $y$, then we can show that this condition is equivalent to the upper quadratic growth condition in Guille-Escuret et al. [21].[3] This observation was first made by Goujaud et al. [18, Theorem 2.6], we include the precise statement and proof in the Appendix B.

The key observation in the next Theorem is that $\phi$ is *always* locally star upper curved, regardless of the curvature (or lack of it) of the function $f$. Moreover, it will inherit additional curvature from $f$. The proof can be found in Appendix A.

**Theorem 1** (Curvature of the Polyak surrogate). *Let $f(x)$ be convex and define $x^\star \in \operatorname{argmin}_x f(x)$. Define $\phi(x) = \frac{1}{2}(f(x) - f(x^\star))^2$. Then, we have*

- *$\phi$ is $\|g_y\|^2$-LSUC around any $y$ for any $g_y \in \partial f(y)$.*

- *If $f$ is $s$-sharp, then $\phi$ has $s^2$-quadratic growth.*

- *If $f$ has $\mu$-quadratic growth and $L$-self bounded, then $\phi$ satisfies a local quadratic growth:*

$$\phi(x) \geq \frac{1}{2} \frac{\mu\|g\|^2}{2L}\|x - x^\star\|^2, \ \forall g \in \partial f(x) \ .$$

This theorem tells us that, regardless of the curvature of $f$, we can always construct the function $\phi$ that is locally curved. It is well-known that for $L$-smooth functions one can use the stepsize $\eta = \frac{1}{L}$ and achieve a rate between $\mathcal{O}(1/T)$ and a linear one, depending on the presence of strong convexity. Here, we show a similar result: GD can use stepsizes that depend on the local star upper curvature in *all* cases. Note however, unlike GD with a constant stepsize and smoothness, we do not have a descent lemma with $\phi$. Indeed this is not possible as it would guarantee a $\mathcal{O}(1/T)$ rate for the last iterate which was shown to be impossible by Goujaud et al. [18] for $QG + (L)$ functions (i.e., L-LUSC functions).

**Lemma 1.** *Let $\phi$ convex and define $x^\star \in \operatorname{argmin}_x \phi(x)$. Assume $\phi$ to be $\lambda_x$-LSUC around any point $x$. Then, using subgradient descent with stepsizes $\eta_t = \frac{1}{\lambda_{x_t}}$ guarantees*

$$\eta_t \left( \phi(x_t) - \phi(x^\star) \right) \leq \frac{1}{2}\|x_t - x^\star\|^2 - \frac{1}{2}\|x_{t+1} - x^\star\|^2 \ . \tag{3}$$

*Summing this inequality over time, we also have*

$$\phi(\bar{x}_T) \sum_{t=1}^{T} \eta_t \leq \sum_{t=1}^{T} \eta_t \phi(x_t) \leq \frac{\|x_1 - x^\star\|^2}{2}, \tag{4}$$

---

[3]A function $f$ satisfies the $\mu$-quadratic growth condition if $f(x) - f(x^\star) \geq \mu/2\|x - x^\star\|^2$.

*where* $\bar{\boldsymbol{x}}_T = \frac{1}{\sum_{t=1}^{T}\eta_t}\boldsymbol{x}_t$ *or* $\bar{\boldsymbol{x}}_T = \text{argmin}_{\boldsymbol{x}\in\{\boldsymbol{x}_1,\ldots,\boldsymbol{x}_T\}}\ \phi(\boldsymbol{x}).$

*Proof.* From the classic one-step analysis of GD [see, e.g., 34], we have

$$\eta_t\langle\boldsymbol{g}_t, \boldsymbol{x}_t - \boldsymbol{x}^\star\rangle = \frac{1}{2}\|\boldsymbol{x}_t - \boldsymbol{x}^\star\|^2 - \frac{1}{2}\|\boldsymbol{x}_{t+1} - \boldsymbol{x}^\star\|^2 + \frac{\eta_t^2}{2}\|\boldsymbol{g}_t\|^2,$$

for any $\boldsymbol{g}_{\boldsymbol{x}_t} \in \partial\phi(\boldsymbol{x}_t)$. Now, we use the fact that $\phi$ is LSUC and the definition of $\eta_t$ we obtain the stated bound. Summing from $t = 1$ to $T$ and discarding the negative term on the right hand side concludes the proof. $\qquad\square$

The above discussion can be summarized in the following theorem.

**Theorem 2.** *The Polyak stepsize in* (1) *is equivalent to subgradient descent on the function $\phi$ in* (2), *when using stepsizes $\eta_t$ equal to the inverse of the local-star-upper curvature of $\phi$ in $\boldsymbol{x}_t$.*

This theorem and Lemma 1 do not give us a rate, however we can immediately observe that if $\sum_t \eta_t = +\infty$ we also have $\phi(\boldsymbol{x}_t) \to 0$, implying convergence of the last iterate (see Remark 6 for more details).

To obtain known rates for the Polyak stepsize, we can use additional assumptions on $f$. However, we want to stress that, differently from prior results, we explicitly get a convergence rate for the surrogate function $\phi$, the function actually minimized by (1). The rates on the original function $f$ are immediate by just taking the square root.

If $f$ is $G$-Lipschitz, then $\sum_{t=1}^{T}\eta_t \geq T/G^2$. Hence, the final rate is $\phi(\bar{\boldsymbol{x}}_T) \leq \frac{G^2}{2T}\|\boldsymbol{x}_1 - \boldsymbol{x}^\star\|^2$. So, the surrogate loss converges as $\mathcal{O}(1/T)$, as expected by a loss with upper curvature.

Now, instead let's assume that the function $f$ is $L$-self-bounded. Using inequalities between harmonic and arithmetic means, we have

$$\frac{1}{\sum_{t=1}^{T}\frac{1}{\|\boldsymbol{g}_{\boldsymbol{x}_t}\|^2}} \leq \frac{1}{T^2}\sum_{t=1}^{T}\|\boldsymbol{g}_{\boldsymbol{x}_t}\|^2 = \frac{1}{T^2}\sum_{t=1}^{T}\frac{\|\boldsymbol{g}_{\boldsymbol{x}_t}\|^4}{\|\boldsymbol{g}_{\boldsymbol{x}_t}\|^2} \leq \frac{1}{T^2}\sum_{t=1}^{T}\frac{8L^2\phi(\boldsymbol{x}_t)}{\|\boldsymbol{g}_{\boldsymbol{x}_t}\|^2} \leq \frac{8L^2}{T^2}\frac{1}{2}\|\boldsymbol{x}_1 - \boldsymbol{x}^\star\|^2,$$

where in the last inequality we use (4). This implies

$$\phi(\bar{\boldsymbol{x}}_T) \leq \frac{1}{2\sum_{t=1}^{T}\eta_t}\|\boldsymbol{x}_1 - \boldsymbol{x}^\star\|^2 = \frac{1}{2\sum_{t=1}^{T}\frac{1}{\|\boldsymbol{g}_{\boldsymbol{x}_t}\|^2}}\|\boldsymbol{x}_1 - \boldsymbol{x}^\star\|^2 \leq \frac{4L^2}{T^2}\|\boldsymbol{x}_1 - \boldsymbol{x}^\star\|^4\ .$$

Similarly, it is equally easy to obtain rates for Hölder-smooth functions, see Theorem 7 for more details.

We can also assume that $f$ is $s$-sharp and $G$-Lipschitz. So, from (3) and the fact that $\phi$ has $s^2$ quadratic growth from Theorem 1, then by Lemma 1 we have

$$\frac{s^2}{G^2}\frac{1}{2}\|\boldsymbol{x}_t - \boldsymbol{x}^\star\|^2 \leq s^2\eta_t\frac{1}{2}\|\boldsymbol{x}_t - \boldsymbol{x}^\star\|^2 \leq \eta_t(\phi(\boldsymbol{x}_t) - \phi(\boldsymbol{x}^\star)) \leq \frac{1}{2}\|\boldsymbol{x}_t - \boldsymbol{x}^\star\|^2 - \frac{1}{2}\|\boldsymbol{x}_{t+1} - \boldsymbol{x}^\star\|^2\ .$$

Using the fact that $\frac{s^2}{G^2} \leq 1$, this immediately gives a linear convergence rate.

## 5 Generalizing the Polyak Stepsize: More Surrogates and Stochastic Setting

We have shown how the Polyak stepsize is just GD on a particular function with stepsizes adapted to the local curvature of the function. In this section, we show that we can construct an entire family of surrogate losses with similar guarantees, while also preparing ourselves for the stochastic setting.

Instead of the function (2), we consider more generally the surrogate

$$\psi(\boldsymbol{x}) = \frac{1}{2}h^2(\boldsymbol{x}),$$

where $h : \mathbb{R}^d \to \mathbb{R}_{\geq 0}$ is convex. As a special case we can recover (2) with $h(\boldsymbol{x}) = f(\boldsymbol{x}) - f^\star$, however, in general we do not need to know $f^\star$. For example we can take $h = (f(\boldsymbol{x}) - a)_+$ for any $a$. Intuitively, the role of $h$ is to transform $f$ into a positive function. We show that $\psi$ generally has an approximate local-star-upper curvature, where the approximation stems from $h$ potentially being strictly positive in $\boldsymbol{x}^\star$.

**Definition 4** (Approximate local-star-upper curvature). *We will say that a function $f$ with minimizer $\boldsymbol{x}^\star$ has $\epsilon$-approximate $\lambda_{\boldsymbol{y}}$-star-upper-curvature around $\boldsymbol{y}$ if there exists $\epsilon$ such that*

$$f(\boldsymbol{x}^\star) - f(\boldsymbol{y}) - \langle \boldsymbol{g}_{\boldsymbol{y}}, \boldsymbol{x}^\star - \boldsymbol{y} \rangle \geq \frac{1}{2\lambda_{\boldsymbol{y}}} \|\boldsymbol{g}_{\boldsymbol{y}}\|_2^2 - \epsilon, \ \forall \boldsymbol{g}_{\boldsymbol{y}} \in \partial f(\boldsymbol{y}) .$$

Since we do not make explicit assumptions $h$ with respect to $f$, we can only hope to achieve convergence to the minimum of $h$ or $\psi$. So, from here onward we denote $\boldsymbol{x}^\star$ as as minizer of $\psi$.

**Lemma 2.** *Let $h : \mathbb{R}^d \to \mathbb{R}_{\geq 0}$ be convex. Define $\psi = \frac{1}{2}h^2$. Then, $\psi$ is $(2\sqrt{\psi(\boldsymbol{x})\psi(\boldsymbol{x}^\star)} - \psi(\boldsymbol{x}^\star))$-approximate $\|\boldsymbol{g}\|$-LSUC for any $\boldsymbol{g} \in \partial h(\boldsymbol{x})$.*

*Proof.* Given that the function $\psi(\boldsymbol{x})$ might not be differentiable, we have to be careful in the calculation of its subgradients. We have

$$\psi(\boldsymbol{y}) = \frac{1}{2}h^2(\boldsymbol{y}) \geq \frac{1}{2}h^2(\boldsymbol{x}) + h(\boldsymbol{x})[h(\boldsymbol{y}) - h(\boldsymbol{x})]$$
$$\geq \frac{1}{2}h^2(\boldsymbol{x}) + h(\boldsymbol{x})\langle \boldsymbol{g}, \boldsymbol{y} - \boldsymbol{x} \rangle = \psi(\boldsymbol{x}) + \langle h(\boldsymbol{x})\boldsymbol{g}, \boldsymbol{y} - \boldsymbol{x} \rangle,$$

where $\boldsymbol{g} \in \partial h(\boldsymbol{x})$ and the first inequality is due to the fact that $\frac{1}{2}(\cdot)^2$ is a convex function. Hence, we see that $\tilde{\boldsymbol{g}} := h(\boldsymbol{x})\boldsymbol{g}$ is a subgradient of $\psi$ in $\boldsymbol{x}$. Hence, for any $\boldsymbol{u} \in \mathbb{R}^d$, we have

$$\psi(\boldsymbol{x}) - \langle \tilde{\boldsymbol{g}}, \boldsymbol{x} - \boldsymbol{u} \rangle + \frac{1}{2\|\boldsymbol{g}\|^2}\|\tilde{\boldsymbol{g}}\|_2^2 = \frac{1}{2}h^2(\boldsymbol{x})) - h(\boldsymbol{x})\langle \boldsymbol{g}, \boldsymbol{x} - \boldsymbol{u} \rangle + \frac{1}{2}h(\boldsymbol{x})^2$$
$$= h(\boldsymbol{x})\left(h(\boldsymbol{x}) - \langle \boldsymbol{g}, \boldsymbol{x} - \boldsymbol{u} \rangle\right)$$
$$= h(\boldsymbol{x})\left(h(\boldsymbol{x}) - h(\boldsymbol{u}) - \langle \boldsymbol{g}, \boldsymbol{x} - \boldsymbol{u} \rangle + h(\boldsymbol{u})\right)$$
$$\leq h(\boldsymbol{x}) \cdot h(\boldsymbol{u}) = 2\sqrt{\psi(\boldsymbol{x})\psi(\boldsymbol{u})},$$

where the inequality is due to the convexity of $h$ and the fact that $h(\boldsymbol{x}) \geq 0$. Setting $\boldsymbol{u} = \boldsymbol{x}^\star$, we have the stated bound. $\square$

With approximate local curvature, a generalization of Lemma 1 is immediate.

**Lemma 3.** *Assume $\psi : \mathbb{R}^d \to \mathbb{R}$ to be $\epsilon_t$-approximately $\lambda_t$-star-upper-curve around $\boldsymbol{x}_t$. Then, for any $\eta_t > 0$, any $\boldsymbol{g}_t \in \partial \psi(\boldsymbol{x}_t)$, and $\boldsymbol{x}_{t+1} = \boldsymbol{x}_t - \eta_t \boldsymbol{g}_t$ we have*

$$\eta_t(\psi(\boldsymbol{x}_t) - \psi(\boldsymbol{x}^\star)) \leq \frac{\|\boldsymbol{x}^\star - \boldsymbol{x}_t\|_2^2}{2} - \frac{\|\boldsymbol{x}^\star - \boldsymbol{x}_{t+1}\|_2^2}{2} + \frac{\eta_t}{2}\left(\eta_t - \frac{1}{\lambda_t}\right)\|\boldsymbol{g}_t\|^2 + \eta_t \epsilon_t .$$

The last two lemmas tell us that the properties of the surrogate functions breaks if $\psi(\boldsymbol{x}^\star) \neq 0$. Hence, we will not be able to prove convergence results, but only that, for example, the suboptimality gap will converge up to a floor that depends on $\psi(\boldsymbol{x}^\star)$. However, in Section 6 we will show that this is not an artifact of the proof. Indeed, we can construct simple one-dimensional functions where the generalized Polyak stepsize does not converge.

## 5.1 Stochastic Approximation Setting

Consider now the case that we are minimizing $F(\boldsymbol{x}) := \mathbb{E}_{\boldsymbol{\xi} \sim D}[f(\boldsymbol{x}, \boldsymbol{\xi})]$, where $f : \mathbb{R}^d \times \mathcal{S} \to \mathbb{R}$, that covers both the stochastic approximation and finite-sum settings. We do not know the distribution $D$, but we assume that we can sample $\boldsymbol{\xi}$ i.i.d. from $D$.

In this setting, we argue that the Polyak stepsize makes sense only in restricted settings. In fact, the interpretation of the Polyak stepsize as minimizing a surrogate function implies that in the stochastic setting we will minimize the function $\mathbb{E}_{\boldsymbol{\xi} \sim D}[\frac{1}{2}h^2(\boldsymbol{x}, \boldsymbol{\xi})]$, where the function $h(\cdot, \boldsymbol{\xi})$ depends on the particular variant of the stochastic Polyak stepsize. It is clear that in general $\operatorname{argmin}_{\boldsymbol{x}} \mathbb{E}_{\boldsymbol{\xi} \sim D}[f(\boldsymbol{x}, \boldsymbol{\xi})]$ can be completely different from $\operatorname{argmin}_{\boldsymbol{x}} \mathbb{E}_{\boldsymbol{\xi} \sim D}[\frac{1}{2}h^2(\boldsymbol{x}, \boldsymbol{\xi})]$.

Here, starting from ALI-G [5] and $\text{SPS}_{\max}$ [29] that use the idea of limiting the stepsizes, we propose a generalized Polyak stepsize algorithm. The proof is in Appendix C.

---

**Algorithm 1** Generalized Polyak Stepsize

---

**Require:** $h : \mathbb{R}^d \times \mathcal{S} \to \mathbb{R}$, $\boldsymbol{x}_1 \in \mathbb{R}^d$
1: **for** $t = 1, \dots, T$ **do**
2:     Sample $\boldsymbol{\xi}_t$ from $D$
3:     Tranform $f(\boldsymbol{x}, \boldsymbol{\xi}_t)$ into $h(\boldsymbol{x}, \boldsymbol{\xi}_t)$
4:     Receive $\boldsymbol{g}_t \in \partial h(\boldsymbol{x}, \boldsymbol{\xi}_t)$
5:     **if** $\boldsymbol{g}_t \neq \boldsymbol{0}$ **then**
6:       $\boldsymbol{x}_{t+1} = \boldsymbol{x}_t - \eta_t h(\boldsymbol{x}_t, \boldsymbol{\xi}_t) \boldsymbol{g}_t$ where $\eta_t = \min\left( \frac{1}{\|\boldsymbol{g}_t\|^2}, \frac{\gamma}{h(\boldsymbol{x}_t, \boldsymbol{\xi}_t)} \right)$
7:     **else**
8:       $\boldsymbol{x}_{t+1} = \boldsymbol{x}_t$
9:     **end if**
10: **end for**

---

**Theorem 3.** *Let* $h : \mathbb{R}^d \times \mathcal{S} \to \mathbb{R}_{\geq 0}$ *be convex in its first argument. Denote by* $H(\boldsymbol{x}) = \mathbb{E}_{\boldsymbol{\xi} \sim D}[h(\boldsymbol{x}, \boldsymbol{\xi})]$. *Then, setting* $\eta_t = \min\left( \frac{1}{\|\boldsymbol{g}_t\|^2}, \frac{\gamma}{h(\boldsymbol{x}_t, \boldsymbol{\xi}_t)} \right)$ *in Algorithm 1, we have*

- *If* $h(\cdot, \boldsymbol{\xi}_t)$ *is L-self bounded, we have*

$$\frac{1}{T} \sum_{t=1}^{T} \min\left( \frac{1}{2L}, \gamma \right) \mathbb{E}[H(\boldsymbol{x}_t)] \leq \frac{\|\boldsymbol{x}_1 - \boldsymbol{x}^\star\|^2}{T} + 2\gamma H(\boldsymbol{x}^\star)$$

*and*

$$\sum_{t=1}^{T} \min\left( \frac{1}{2L}, \gamma \right) \mathbb{E}[H(\boldsymbol{x}_t)] - \gamma \sum_{t=1}^{T} H(\boldsymbol{x}^\star) \leq \frac{1}{2} \|\boldsymbol{x}_1 - \boldsymbol{x}^\star\|^2 + \frac{1}{2} \sum_{t=1}^{T} \gamma^2 \mathbb{E}[\|\boldsymbol{g}_t\|^2] .$$

- *If* $h(\cdot, \boldsymbol{\xi}_t)$ *is G-Lipschitz, then we have*

$$\frac{1}{T} \sum_{t=1}^{T} \mathbb{E}[H(\boldsymbol{x}_t)] \leq \frac{\|\boldsymbol{x}_1 - \boldsymbol{x}^\star\|^2}{\gamma T} + 2 H(\boldsymbol{x}^\star) + \frac{G\|\boldsymbol{x}_1 - \boldsymbol{x}^\star\|}{\sqrt{T}} + G\sqrt{2\gamma H(\boldsymbol{x}^\star)} .$$

- *If* $h(\cdot, \boldsymbol{\xi})$ *is L-self-bounded and* $H(\boldsymbol{x})$ *has $\mu$-quadratic growth, then*

$$\mathbb{E}\left[ \|\boldsymbol{x}_{T+1} - \boldsymbol{x}^\star\|^2 \right] \leq \mathbb{E}\left[ \|\boldsymbol{x}_1 - \boldsymbol{x}^\star\|^2 \right] a^{T+1} + b \frac{1 - a^{T+1}}{1 - a} H(\boldsymbol{x}^\star),$$

*where* $a = \frac{\mu}{2} \min\left( \frac{1}{2L}, \gamma \right)$ *and* $b = 2\gamma - \min\left( \frac{1}{2L}, \gamma \right)$.

Choosing the function $h$, the above theorem covers and extends a number of results in previous papers, for example:

- SPS$_{\max}$ [29]: $h(\boldsymbol{x}, \boldsymbol{\xi}) = f(\boldsymbol{x}, \boldsymbol{\xi}) - \inf_{\boldsymbol{x}} f(\boldsymbol{x}, \boldsymbol{\xi})$, so $H(\boldsymbol{x}^\star) = \mathbb{E}[f(\boldsymbol{x}^\star, \boldsymbol{\xi}) - \inf_{\boldsymbol{x}} f(\boldsymbol{x}, \boldsymbol{\xi})]$.
- SPS$_{\max}^\ell$ [35]: $h(\boldsymbol{x}, \boldsymbol{\xi}) = f(\boldsymbol{x}, \boldsymbol{\xi}) - q(\boldsymbol{\xi})$, where $q(\boldsymbol{\xi})$ is a lower bound to $\inf_{\boldsymbol{x}} f(\boldsymbol{x}, \boldsymbol{\xi})$. In this case, $H(\boldsymbol{x}^\star) = \mathbb{E}[f(\boldsymbol{x}^\star, \boldsymbol{\xi}) - q(\boldsymbol{\xi})]$.
- SPS$_+$ [15]: $h(\boldsymbol{x}, \boldsymbol{\xi}) = (f(\boldsymbol{x}, \boldsymbol{\xi}) - f(\boldsymbol{x}^\star, \boldsymbol{\xi}))_+$. In this case, $H(\boldsymbol{x}^\star) = 0$ so we can also safely set $\gamma = \infty$. Moreover, $H(\boldsymbol{x}) \geq F(\boldsymbol{x}) - F(\boldsymbol{x}^\star)$, hence any bound on $H(\boldsymbol{x})$ translates to a bound on the suboptimality gap.

**Remark 1.** *If* $h(\boldsymbol{x}^\star, \boldsymbol{\xi}_t) = 0$ *for all t, then* $\|\boldsymbol{x}_{t+1} - \boldsymbol{x}^\star\| \leq \|\boldsymbol{x}_t - \boldsymbol{x}^\star\|$. *Hence, in this case we only need to consider all the properties of h in the bounded domain* $\{\boldsymbol{x} \in \mathbb{R}^d : \|\boldsymbol{x} - \boldsymbol{x}^\star\| \leq \|\boldsymbol{x}_1 - \boldsymbol{x}^\star\|\}$. *This is well-known via the subgradient projector perspective, as* $\boldsymbol{x}_t$ *is guaranteed to approach the set* $\bigcap_{\boldsymbol{\xi}} \{\boldsymbol{x} : h(\boldsymbol{x}, \boldsymbol{\xi}) = 0\}$ *at each iteration if it is non-empty. This property was observed in Gower et al. [20] for SPS$_+$ [15], but here it holds more generally. For example,* $h(\boldsymbol{x}, \boldsymbol{\xi}) = (f(\boldsymbol{x}, \boldsymbol{\xi}) - a)_+$, *where* $a \geq \sup_{\xi} f(\boldsymbol{x}^\star, \boldsymbol{\xi})$, *would also have no neighbourhood of convergence and satisfies the assumption of the theorem.*

**Remark 2.** *The above theorem also applies to the case where some of the $f(\cdot, \xi)$ are non-convex functions, while still guaranteeing the convergence to the global optimum of $F$. For example, consider $F(x) = 0.5f_1(x) + 0.5f_2(x)$, where $f_1 = -|x|$ (non-convex) and $f_2 = 2|x|$. We have that $F(x) = \frac{1}{2}|x|$ so $\boldsymbol{x}^\star = 0$. Now, choose $h_1(x) = \max(f_1(x) - f_1(x^\star), 0) = 0$ and $h_2(x) = \max(f_2(x) - f_2(x^\star), 0) = 2|x|$. Hence, the hypotheses of the theorem are verified. Moreover, $H(x) = 0.5h_1(x) + 0.5h_2(x) \geq F(x)$ and $H(x^\star) = 0$, so the theorem implies a convergence rate for the minimization of $F(x) - F(x^\star)$ by using $\mathtt{SPS}_+$.*

**Remark 3.** *In the proof of Theorem 3, if one stops before taking expectations, one obtains a* regret guarantee *on a sequence of arbitrary losses $h(\boldsymbol{x}, \boldsymbol{\xi}_t)$. Such regret scales as the sum of the loss in $\boldsymbol{x}^\star$. This is exactly the $L^\star$ bound that we mentioned in Section 2. Indeed, this kind of updates and guarantees were already obtained in the online learning literature for the special case of linear predictors by the Passive-Aggressive family of algorithms [11].*

Besides covering a number of previous algorithmic variants, we also extend the previous known guarantees. In particular, Loizou et al. [29] only studied SPS in the non-smooth setting but did not include $\mathtt{SPS}_{\max}$.[4] Theorem 3 shows for the first time that $\mathtt{SPS}_{\max}$ is adaptive to the entire range of upper curvature of the function, from Lipschitz to smooth functions. In Appendix D we also show additional results. Moreover, the second result in the smooth case is new, and it allows to recover the SGD guarantee on $H$ when $\gamma$ is sufficiently small. For example, we include a precise statement for $\mathtt{SPS}_+$ when $f(\boldsymbol{x}, \boldsymbol{\xi})$ is $L$-self-bounded, that recovers the guarantee in Gower et al. [20, Corollary 2.3].

## 6 Neighbourhood of Convergence and Instability of the Polyak Stepsize

In Section 5 we demonstrate that a generalized version of the Polyak stepsize and existing variants can be viewed as GD on a function with approximate local curvature, with convergence to a neighbourhood of the optimal solution. This neighbourhood of convergence appears in our analysis just like in all existing variants, therefore *suggesting* it is unavoidable if

$$H(\boldsymbol{x}) = \mathbb{E}_{\boldsymbol{\xi} \sim D}[h(\boldsymbol{x}, \boldsymbol{\xi})] > 0 \text{ for all } \boldsymbol{x} . \tag{5}$$

In this section, we demonstrate that this neighbourhood of convergence is not an artifact of the analysis and indeed cannot be avoided in general. We also show that the positivity condition (5) can fundamentally change the dynamics of Algorithm 1, even in the deterministic setting, thus posing a challenge that is not just associated with interpolation.

Condition (5) occurs with SPS [29] without interpolation, or in the deterministic setting when the optimal value is underestimated, $h(\boldsymbol{x}) = f(\boldsymbol{x}) - c$ where $f^\star > c$. Convergence under this condition was first studied in the deterministic case in Polyak's original paper [36], where it is shown that if $\inf_x h(x) = h^\star > 0$ then $\lim_{t \to \infty} \min_{1 \leq s \leq t} h(\boldsymbol{x}_t) - h^\star \leq h^\star$. That is, the best iterate eventually enters a neighbourhood of the minima where the size of the neighbourhood is dependent on how much $h^\star$ is underestimated by 0. In the stochastic case, convergence of the average iterate to a neighbourhood when understimating the minimum was also studied by Orvieto et al. [35] under $\mathtt{SPS}_{\max}^\ell$. However, we demonstrate the consequence of condition (5) is far greater than existing results suggest, with instability of fixed points, potential cycles, lower bounds in the sub-optimality gap, and lack of convergence regardless of initialization.

**Deterministic Setting.** We first demonstrate that in the deterministic setting, for different classes of $h$, if $h^\star = \min_x h(\boldsymbol{x}) > 0$, then the fixed points of

$$\boldsymbol{x}_{t+1} = T(\boldsymbol{x}_t) := \begin{cases} \boldsymbol{x}_t - \frac{h(\boldsymbol{x}_t)}{\|\boldsymbol{g}_t\|^2}\boldsymbol{g}_t, & \text{if } \boldsymbol{g}_t \in \partial h(\boldsymbol{x}_t), 0 \notin \partial h(\boldsymbol{x}_t), \\ \boldsymbol{x}_t, & \text{otherwise} \end{cases} \tag{6}$$

are unstable. Intuitively, this can be explained via our surrogate function view: In fact, denoting the local curvature constant of the surrogate $\frac{1}{2}(h(\boldsymbol{x}) - h^\star)^2$ around $\boldsymbol{x}_t$ as $\lambda_t$, we see that the stepsize $\eta_t$ in update (6) can be equivalently written as

$$\eta_t = \left(\frac{h(\boldsymbol{x}_t)}{h(\boldsymbol{x}_t) - h^\star}\right)\frac{1}{\lambda_t}. \tag{7}$$

---

[4]Although Loizou et al. [29] state that SPS analysis can be readily extended to $\mathtt{SPS}_{\max}$ this does not seem to be the case due to the non-convexity of the min function, as demonstrated by our different proof technique in Appendix C.

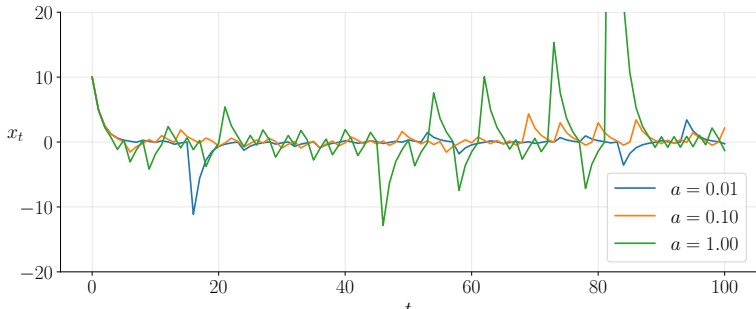

Figure 2: Trajectories under $T$ (6) for $h(x) = \frac{x^2}{2} + a$ with an unstable fixed point at $x^\star = 0$. Lack of convergence is observed for different values of $a$ as predicted by Proposition 3.

If $h$ is Lipschitz or self-bounded then $\eta_t \to +\infty$ as $\boldsymbol{x}_t \to \boldsymbol{x}^\star$. Therefore, if $h$ possesses curvature, then $\boldsymbol{x}_{t+1}$ may move further away from $\boldsymbol{x}^\star$ within a neighborhood of $\boldsymbol{x}^\star$. Indeed, in Proposition 1 we show that for all self-bounded functions with a quadratic growth condition, the fixed point of $T$ in (6) is unstable. A similar result can also be shown if $h$ is $L$-Lipschitz and has a sharp mininum (see Proposition 6 in the Appendix).

**Proposition 1** (Unstable fixed point)**.** *Suppose $h$ is convex, strictly positive, $L$-self-bounded, and satisfies the quadratic growth condition $h(\boldsymbol{x}) - h^\star \geq \frac{\mu}{2}\|\boldsymbol{x} - \boldsymbol{x}^\star\|^2$, where $\boldsymbol{x}^\star = \arg\min_{\boldsymbol{x}} h(\boldsymbol{x})$ is the only fixed point of $T$, defined in (6). Then, for any point $\boldsymbol{x} \in S = \{\boldsymbol{y} : \boldsymbol{y} \neq \boldsymbol{x}^\star, h(\boldsymbol{y}) - h^\star < h^\star \frac{\mu}{8L-\mu}\}$ we have*

$$\|T(\boldsymbol{x}) - \boldsymbol{x}^\star\| > \|\boldsymbol{x} - \boldsymbol{x}^\star\|\ .$$

Note that this reinforces the need to clip the stepsize as proposed in ALI-G and $\text{SPS}_{\max}$. However, clipping will not remove this behaviour unless the maximum value is taken to be small enough. In Proposition 8 we show there is always a subregion where the stepsize is bounded, and this subregion can be made arbitrarily large within the unstable region; therefore, if the clipped value is too large instability is unavoidable.

The importance of $h > 0$ in update (6) has also been studied in Bauschke et al. [3], where they demonstrate with examples that $T$ can fail to be quasi-firmly non-expansive if $h^\star > 0$. Propositions 1, and 6 provide extra insight on this phenomenon as they automatically prove $T$ cannot be quasi-firmly non-expansive and therefore we have the following remark.

**Remark 4.** *If $h > 0$, and either of the following conditions hold:*

- *$h$ is convex, self-bounded, and satisfies the quadratic growth condition,*

- *$h$ is convex, Lipschitz, and has a sharp minimum,*

*then $T$ from (6) is not quasi-firmly non-expansive.*

While Propositions 1 and 6 establish that minima can be unstable, this property may not fully describe the dynamics of update (6). In fact, instability can admit convergence in the average iterate or last iterate if the local critical neighborhood is skipped. So, in Proposition 2 we provide an example of a function $h$, which satisfies the assumptions of Proposition 1, where the iterates *cycle and never reach the minimum in best iterate or on average*.

**Proposition 2** (Cycling and failure to converge)**.** *There exists a strictly positive smooth and strongly convex function $h$, and an initial point $\boldsymbol{x}_1$ such that iterates from update (6) cycle and satisfy the inequality $h(\frac{1}{t}\sum_{i=1}^{t} \boldsymbol{x}_i) - h^\star \geq \delta > 0$ for all $t$.*

Note that since the cycle in Proposition 2 consists of a finite number of points, clipping will not necessarily remove this behaviour (e.g. if the clipped value is taken to be larger than any of the seen stepsizes). In Proposition 2, a specific initialization was chosen to construct a cycle that would not converge. However, in Proposition 3 we show that for 1-d quadratics the lack of convergence is true for all initializations and values of $h^\star$ up to a set of measure zero.

**Proposition 3** (The set of good initializations can have measure zero). *Let $h : \mathbb{R} \to \mathbb{R}$, defined as $h(x) = \frac{x^2}{2} + a$ for $a > 0$, where $x_{t+1} = x_t - \frac{h(x_t)}{\|\nabla h(x_t)\|^2} \nabla h(x_t)$, and $\boldsymbol{x}_1$ is randomly initialized. Then, $P\{\lim_{t\to\infty} x_t = x^\star\} = 0$. In other words, the set of initializations that can converge to the optimal solution has measure zero.*

*Proof.* Note $h$ is 1-smooth and 1-strongly convex and therefore satisfies the conditions of Proposition 1 with $\mu = L = 1$ and an unstable unique fixed point. Let $T$ be such that $x_{t+1} = T(x_t)$. $T(x) = x \left( \frac{1}{2} - \frac{a}{x^2} \right)$ if $x \neq 0$ and $0$ otherwise. With inverse $T^{-1}(S) = \{x \pm \sqrt{x^2 + 2a} : x \in S\}$. Therefore, $T^{-k}(\{x^\star\})$ has at most $2^k$ points which has measure zero for all $k$. By Lemma 5 in Appendix E, the result follows. $\qquad\square$

**Remark 5.** *Note that, by Lemma 5, Proposition 3 can be extended much more generally if $T$ from (6) is shown to satisfy the Lusin ($N^{-1}$) property (see Definition 7) [23, Definition 4.12].*

In the stochastic case with SPS, condition (5) is due to lack of interpolation, and Orvieto et al. [35] show that it can change the expected fixed point. In contrast, in the deterministic setting we have shown lack of convergence despite the fixed point being $\boldsymbol{x}^\star$. Therefore the issue here stems from the instability of the method due to the underestimation of $h^\star$ and not the bias of the expected update.

**Stochastic Setting.** In the stochastic setting, the positivity condition (5) can occur despite $\min_{\boldsymbol{x}} h(\boldsymbol{x}, \boldsymbol{\xi}) = 0$, such as in SPS ($h(\boldsymbol{x}, \boldsymbol{\xi}) = f(\boldsymbol{x}) - \min_{\boldsymbol{x}} f(\boldsymbol{x}, \boldsymbol{\xi})$) without interpolation. Orvieto et al. [35] demonstrate that without interpolation SPS can fail to converge in a 1-d quadratic and has an expected fixed point different than $\min_{\boldsymbol{x}} F(\boldsymbol{x}) = \min_{\boldsymbol{x}} \mathbb{E}_{\boldsymbol{\xi} \sim D}[f(\boldsymbol{x}, \boldsymbol{\xi})]$. Similarly to the deterministic setting, we can show that SPS can have a random walk between a finite number of points.

**Proposition 4** (Failure to converge). *There exist $f_1$ and $f_2$ quadratic 1-d functions and a starting point $x_1$ such that SPS on $F(x) = 0.5(f_1(x) + f_2(x))$ satisfies*
$$\mathbb{E}[F(x_t)] - \min_x F(x) \geq 2/3, \ \forall t \ .$$

*Proof.* Let $f_1 = x^2 + 2x + 5$ and $f_2(x) = 2x^2 - 4x + 10$. Let's start from $x_1 = 1$ where $F(x_1) = 8$. If we draw $f_1$, $x_2 = -1$, while if we draw $f_2$ then $x_2 = 1$ because $f_2'(1) = 0$. Hence, $\mathbb{E}[F(x_2)] = 0.5 \frac{f_1(1) + f_2(1)}{2} + 0.5 \frac{f_1(-1) + f_2(-1)}{2} = 9$. Iterating, we have that $x_3$ has equal probability to be equal to 1 and -1. Hence, again we have $\mathbb{E}[F(x_3)] = 9$. So, we have that this holds for any $t$. Moreover, we have that $\min_x F(x) = 44/6$. $\qquad\square$

## 7 Discussion and Limitations

We have shown that the design, properties, and failure of the (variants) of the Polyak stepsize can be easily derived through the lens of the minimization of a surrogate objective function. This framework also provides a new and natural explanation on the adaptivity of the stepsize via the local curvature of the surrogate. We believe this framework has the promise to design new variants, by simply designing surrogate functions with the required properties. Furthermore, with our perspective we have provided new insight on the challenge of controlling neighbourhoods of convergence that often appear in variants of the Polyak stepsize. We demonstrate that this neighbourhood is unavoidable and a fundamental issue causing instability. Moreover, we show that this issue is not due to the lack of interpolation, as commonly believed, but instead because the minimum of the surrogate loss is not zero more generally.

The limitations of our framework include the assumption of convex $h$ in the generalized surrogate that must be assumed apriori. It is unclear if this framework can be extended to the more general case of noncovex surrogate functions. The class of such surrogates that admit fast rates and tight neighbourhoods of convergence remains an open question that we leave to future work.

## Acknowledgments

We acknowledge the use of Gemini 2.5 in developing the proof of Proposition 2. We also thank Mehdi Inane Ahmed for helpful discussions. Ryan D'Orazio's work is funded by Ioannis Mitliagkas' CIFAR chair.

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

## A  Proofs for the Surrogate $\phi$

**Theorem 1** (Curvature of the Polyak surrogate). *Let $f(x)$ be convex and define $\boldsymbol{x}^\star \in \arg\min_{\boldsymbol{x}} f(\boldsymbol{x})$. Define $\phi(\boldsymbol{x}) = \frac{1}{2}(f(\boldsymbol{x}) - f(\boldsymbol{x}^\star))^2$. Then, we have*

- *$\phi$ is $\|\boldsymbol{g_y}\|^2$-LSUC around any $\boldsymbol{y}$ for any $\boldsymbol{g_y} \in \partial f(\boldsymbol{y})$.*

- *If $f$ is $s$-sharp, then $\phi$ has $s^2$-quadratic growth.*

- *If $f$ has $\mu$-quadratic growth and $L$-self bounded, then $\phi$ satisfies a local quadratic growth:*

$$\phi(\boldsymbol{x}) \geq \frac{1}{2} \frac{\mu \|\boldsymbol{g}\|^2}{2L} \|\boldsymbol{x} - \boldsymbol{x}^\star\|^2, \ \forall \boldsymbol{g} \in \partial f(\boldsymbol{x}) .$$

*Proof.*

$$\phi(\boldsymbol{y}) - \phi(\boldsymbol{x}^\star) - \langle \nabla \phi(\boldsymbol{y}), \boldsymbol{x} - \boldsymbol{x}^\star \rangle + \frac{1}{2\|\boldsymbol{g_y}\|^2} \|\nabla \phi(\boldsymbol{y})\|_2^2$$

$$= \frac{1}{2}(f(\boldsymbol{y}) - f(\boldsymbol{x}^\star))^2 - (f(\boldsymbol{y}) - f(\boldsymbol{x}^\star))\langle \boldsymbol{g_y}, \boldsymbol{y} - \boldsymbol{x}^\star \rangle + \frac{1}{2\|\boldsymbol{g_y}\|^2} \|\nabla \phi(\boldsymbol{y})\|^2$$

$$= \frac{1}{2}(f(\boldsymbol{y}) - f(\boldsymbol{x}^\star))^2 - (f(\boldsymbol{y}) - f(\boldsymbol{x}^\star))\langle \boldsymbol{g_y}, \boldsymbol{y} - \boldsymbol{x}^\star \rangle + \frac{(f(\boldsymbol{y}) - f(\boldsymbol{x}^\star))^2}{2}$$

$$= (f(\boldsymbol{y}) - f(\boldsymbol{x}^\star))\left(f(\boldsymbol{y}) - f(\boldsymbol{x}^\star) - \langle \boldsymbol{g_y}, \boldsymbol{y} - \boldsymbol{x}^\star \rangle\right)$$

$$\leq 0,$$

where the inequality is due to the convexity of $f$ and the fact that $f(\boldsymbol{y}) - f(\boldsymbol{x}^\star) \geq 0$.

For the second property, we have

$$f(\boldsymbol{x}) - f(\boldsymbol{x}^\star) \geq \alpha \|\boldsymbol{x} - \boldsymbol{x}^\star\| \Rightarrow \phi(\boldsymbol{x}) - \phi(\boldsymbol{x}^\star) \geq \frac{\alpha^2}{2} \|\boldsymbol{x} - \boldsymbol{x}^\star\|^2 .$$

For the third property we have

$$\phi(\boldsymbol{x}) = \frac{1}{2}(f(\boldsymbol{x}) - f(\boldsymbol{x}^\star))^2 \geq \frac{\|\boldsymbol{g}\|^2}{2L}(f(\boldsymbol{x}) - f(\boldsymbol{x}^\star)) \geq \frac{1}{2} \frac{\mu \|\boldsymbol{g}\|^2}{2L} \|\boldsymbol{x} - \boldsymbol{x}^\star\|^2.$$

$\square$

**Remark 6.** *Convergence of the last iterate follows from a classic argument with Fejér monotone sequences. From Lemma 1 we have that the distance to any solution is decreasing $\|\boldsymbol{x}_{t+1} - \boldsymbol{x}^\star\| \leq \|\boldsymbol{x}_t - \boldsymbol{x}^\star\|$ for any minimizer $\boldsymbol{x}^\star$ of $\phi$, that is, $\{\boldsymbol{x}_t\}_{t \geq 0}$ is a Fejér monotone sequence with respect to the solution set. Since we have $\phi(\boldsymbol{x}_t) \to 0$ and $\phi$ is continuous then for every limit point $\boldsymbol{x}'$ of the sequence it also holds that $\phi(\boldsymbol{x}') = 0$ implying $\boldsymbol{x}'$ is also a minimizer of $\phi$. Therefore we can use the fact that if $\{\boldsymbol{x}_t\}_{t \geq 0}$ is Fejér monotone with respect to the solution set and the set contains all the limit points of the sequence then the sequence must converge to a point in the solution set (see Theorem 8.16 in Beck [4]).*

## B  Relationship between Star Upper Curvature and Upper Quadratic Growth

For a function $f$, denote by $\mathcal{X}^\star := \{\boldsymbol{x} : f(\boldsymbol{x}) = \min_{\boldsymbol{x}} f(\boldsymbol{x})\}$. In Guille-Escuret et al. [21], they define the following function class.

**Definition 5.** *A function $f$ is $L$-quadratically upper bounded (denoted $L$-$QG^+$) if for all $\boldsymbol{x} \in \mathbb{R}^d$:*

$$f(\boldsymbol{x}) - f^\star \leq \frac{L}{2} \min_{\boldsymbol{x}^\star \in \mathcal{X}^\star} \|\boldsymbol{x} - \boldsymbol{x}^\star\|_2^2 .$$

We now show that convex $L$-$QG^+$ are globally $L$-star upper curved, while the other direction is true for the local version of the two definitions.

**Theorem 5.** *Let $f$ be a convex $L$-$QG^+$ function, then $f$ is globally $L$-star upper curved. On the other hand, let $f$ be $L_x$-LSUC, then for all $x$ we have*

$$f(\boldsymbol{x}) - f^\star \leq \frac{L_{\boldsymbol{x}}}{2} \min_{\boldsymbol{x}^\star \in \mathcal{X}^\star} \|\boldsymbol{x} - \boldsymbol{x}^\star\|_2^2 \,.$$

*Proof.* Assume that $f$ is $L$-$QG^+$. Then, we have

$$f(\boldsymbol{y}) + \langle \boldsymbol{g}, \boldsymbol{x} - \boldsymbol{y} \rangle \leq f(\boldsymbol{x}) \leq f^\star + \frac{L}{2} \min_{\boldsymbol{x}' \in \mathcal{X}^\star} \|\boldsymbol{x} - \boldsymbol{x}'\|_2^2,$$

where the first inequality is due to convexity and $\boldsymbol{g} \in \partial f(\boldsymbol{y})$. Now, set $\boldsymbol{x} = \boldsymbol{x}^\star + \frac{1}{L}\boldsymbol{g}$ for any $\boldsymbol{x}^\star \in \mathcal{X}^\star$, to have

$$f(\boldsymbol{y}) - f^\star \leq -\langle \boldsymbol{g}, \boldsymbol{x}^\star + \frac{1}{L}\boldsymbol{g} - \boldsymbol{y} \rangle + \frac{L}{2} \min_{\boldsymbol{x}' \in \mathcal{X}^\star} \left\| \boldsymbol{x}^\star + \frac{1}{L}\boldsymbol{g} - \boldsymbol{x}' \right\|_2^2 \leq -\langle \boldsymbol{g}, \boldsymbol{x}^\star + \frac{1}{L}\boldsymbol{g} - \boldsymbol{y} \rangle + \frac{\|\boldsymbol{g}\|_2^2}{2L}$$

$$= \langle \boldsymbol{g}, \boldsymbol{y} - \boldsymbol{x}^\star \rangle - \frac{1}{2L}\|\boldsymbol{g}\|_2^2 \,.$$

Now, assume that $f$ is $\lambda_{\boldsymbol{x}}$-LSUC and set $\boldsymbol{g} \in \partial f(\boldsymbol{x})$. For any $\boldsymbol{x}^\star \in \operatorname{argmin}_{\boldsymbol{x}} f(\boldsymbol{x})$, using Cauchy-Schwarz's inequality, we have

$$f(\boldsymbol{x}) - f(\boldsymbol{x}^\star) \leq \langle \boldsymbol{g}, \boldsymbol{x} - \boldsymbol{x}^\star \rangle - \frac{1}{2\lambda_{\boldsymbol{x}}}\|\boldsymbol{g}\|_2^2 \leq \|\boldsymbol{g}\|_2\|\boldsymbol{x} - \boldsymbol{x}^\star\|_2 - \frac{1}{2\lambda_{\boldsymbol{x}}}\|\boldsymbol{g}\|_2^2 \leq \frac{\lambda_{\boldsymbol{x}}}{2}\|\boldsymbol{x} - \boldsymbol{x}^\star\|_2^2 \,.$$

Given that this holds for all $\boldsymbol{x}^\star \in \mathcal{X}^\star$, it implies

$$f(\boldsymbol{x}) - f^\star \leq \frac{\lambda_{\boldsymbol{x}}}{2} \min_{\boldsymbol{x}' \in \mathcal{X}^\star} \|\boldsymbol{x} - \boldsymbol{x}'\|_2^2 \,. \qquad \square$$

## C   Proofs for the Stochastic Surrogate $\psi$

**Theorem 3.** *Let $h : \mathbb{R}^d \times \mathcal{S} \to \mathbb{R}_{\geq 0}$ be convex. Denote by $H(\boldsymbol{x}) = \mathbb{E}_{\boldsymbol{\xi} \sim D}[h(\boldsymbol{x}, \boldsymbol{\xi})]$. Then, setting $\eta_t = \min\left(\frac{1}{\|\boldsymbol{g}_t\|^2}, \frac{\gamma}{h(\boldsymbol{x}_t, \boldsymbol{\xi}_t)}\right)$ in Algorithm 1, we have*

- *If $h(\cdot, \boldsymbol{\xi}_t)$ is $L$-self bounded, we have*

$$\frac{1}{T} \sum_{t=1}^{T} \min\left(\frac{1}{2L}, \gamma\right) \mathbb{E}[H(\boldsymbol{x}_t)] \leq \frac{\|\boldsymbol{x}_1 - \boldsymbol{x}^\star\|^2}{T} + 2\gamma H(\boldsymbol{x}^\star)$$

  *and*

$$\sum_{t=1}^{T} \min\left(\frac{1}{2L}, \gamma\right) \mathbb{E}[H(\boldsymbol{x}_t)] \leq \frac{1}{2}\|\boldsymbol{x}_1 - \boldsymbol{x}^\star\|^2 + \frac{1}{2}\sum_{t=1}^{T} \gamma^2 \mathbb{E}[\|\boldsymbol{g}_t\|^2] + \gamma \sum_{t=1}^{T} H(\boldsymbol{x}^\star) \,.$$

- *If $h(\cdot, \boldsymbol{\xi}_t)$ is $G$-Lipschitz, then we have*

$$\frac{1}{T} \sum_{t=1}^{T} \mathbb{E}[H(\boldsymbol{x}_t)] \leq \frac{\|\boldsymbol{x}_1 - \boldsymbol{x}^\star\|^2}{\gamma T} + 2H(\boldsymbol{x}^\star) + \frac{G\|\boldsymbol{x}_1 - \boldsymbol{x}^\star\|}{\sqrt{T}} + G\sqrt{2\gamma H(\boldsymbol{x}^\star)} \,.$$

- *If $h(\cdot, \boldsymbol{\xi})$ is $L$-self-bounded and $H(\boldsymbol{x})$ has $\mu$-quadratic growth, then*

$$\mathbb{E}\left[\|\boldsymbol{x}_{T+1} - \boldsymbol{x}^\star\|^2\right] \leq \mathbb{E}\left[\|\boldsymbol{x}_1 - \boldsymbol{x}^\star\|^2\right] a^{T+1} + b\frac{1 - a^{T+1}}{1 - a} H(\boldsymbol{x}^\star),$$

  *where $a = \frac{\mu}{2}\min\left(\frac{1}{2L}, \gamma\right)$ and $b = 2\gamma - \min\left(\frac{1}{2L}, \gamma\right)$.*

*Proof.* For simplicity, denote by $h_t(\boldsymbol{x}) = h(\boldsymbol{x}, \xi_t)$.

From the Lemma 3, we have

$$\sum_{t=1}^{T} \eta_t \frac{1}{2} h_t^2(\boldsymbol{x}_t) \leq \frac{1}{2}\|\boldsymbol{x}_1 - \boldsymbol{x}^\star\|^2 + \sum_{t=1}^{T} \frac{\eta_t}{2}\left(\eta_t - \frac{1}{\|\boldsymbol{g}_t\|^2}\right)\|\boldsymbol{g}_t\|^2 h_t^2(\boldsymbol{x}_t) + \sum_{t=1}^{T} \eta_t h_t(\boldsymbol{x}_t) h_t(\boldsymbol{x}^\star) .$$

For the last term in the r.h.s., we have

$$\eta_t h_t(\boldsymbol{x}_t) h_t(\boldsymbol{x}^\star) = \min\left(\frac{1}{\|\boldsymbol{g}_t\|^2}, \frac{\gamma}{h_t(\boldsymbol{x}_t)}\right) h_t(\boldsymbol{x}_t) h_t(\boldsymbol{x}^\star) \leq \gamma h_t(\boldsymbol{x}^\star) .$$

Observe that if $h_t$ is $L$-self bounded then $\|g_t\|^2 \leq 2L(h_t(\boldsymbol{x}) - \inf_{\boldsymbol{x}} h_t(\boldsymbol{x})) \leq 2L h_t(\boldsymbol{x})$. Therefore, we have

$$\min\left(\frac{1}{2L}, \gamma\right) h_t(\boldsymbol{x}_t) \leq \min\left(\frac{h_t(\boldsymbol{x}_t)}{\|\boldsymbol{g}_t\|^2}, \gamma\right) h_t(\boldsymbol{x}_t) = \eta_t h_t^2(\boldsymbol{x}_t) .$$

Now, since $\eta_t \leq 1/\|\boldsymbol{g}_t\|^2$ the second term on the r.h.s can be discarded because it's negative. Taking expectations, we have the first stated bound.

For the second result, bring on the l.h.s. the terms $\frac{\eta_t}{2} h_t^2(\boldsymbol{x}_t)$. Taking expectations, we have the stated bound.

For the third result, first of all observe that for any $a, b > 0$ we have

$$\min(a, b) = \frac{1}{\max(1/a, 1/b)} \geq \frac{1}{1/a + 1/b} .$$

Hence, we have

$$\eta_t h_t^2(\boldsymbol{x}_t) = \min\left(\frac{h_t(\boldsymbol{x}_t)}{\|\boldsymbol{g}_t\|^2}, \gamma\right) h_t(\boldsymbol{x}_t) \geq \gamma \frac{h_t^2(\boldsymbol{x}_t)}{h_t(\boldsymbol{x}_t) + \gamma \|\boldsymbol{g}_t\|^2} \geq \gamma \frac{h_t^2(\boldsymbol{x}_t)}{h_t(\boldsymbol{x}_t) + \gamma G^2} .$$

Now, observe that the function $B(x) = \frac{x^2}{x + \gamma G^2}$ is convex $x \geq 0$, because $B''(x) = \frac{2\gamma^2 G^4}{(\gamma G^2 + x)^3}$. So, summing over time and using Jensen's inequality, we have

$$B\left(\frac{1}{T}\sum_{t=1}^{T} h_t(\boldsymbol{x}_t)\right) \leq \frac{1}{T}\sum_{t=1}^{T} B(h_t(\boldsymbol{x}_t)) = \frac{1}{T}\sum_{t=1}^{T} \frac{h_t^2(\boldsymbol{x}_t)}{h_t(\boldsymbol{x}_t) + \gamma G^2} \leq \frac{\|\boldsymbol{x}_1 - \boldsymbol{x}^\star\|^2}{\gamma T} + \frac{2}{T}\sum_{t=1}^{T} h_t(\boldsymbol{x}^\star) .$$

Note that $B^{-1}(x) = \frac{x + \sqrt{x^2 + 4x\gamma G^2}}{2} \leq x + G\sqrt{x\gamma}$[5], that is an increasing concave function for $x \geq 0$. So, inverting $B$ and taking expectation, we have

$$\frac{1}{T}\sum_{t=1}^{T} \mathbb{E}[H(\boldsymbol{x}_t)] \leq \mathbb{E}\left[B^{-1}\left(\frac{\|\boldsymbol{x}_1 - \boldsymbol{x}^\star\|^2}{\gamma T} + \frac{2}{T}\sum_{t=1}^{T} h_t(\boldsymbol{x}^\star)\right)\right]$$

$$\leq \frac{\|\boldsymbol{x}_1 - \boldsymbol{x}^\star\|^2}{\gamma T} + 2H(\boldsymbol{x}^\star) + \frac{G\|\boldsymbol{x}_1 - \boldsymbol{x}^\star\|}{\sqrt{T}} + G\sqrt{2\gamma H(\boldsymbol{x}^\star)} .$$

For the smooth and quadratic growth case, we have

$$\min\left(\frac{1}{2L}, \gamma\right) h_t(\boldsymbol{x}_t) \leq \eta_t h_t^2(\boldsymbol{x}_t)$$

$$\leq \|\boldsymbol{x}_t - \boldsymbol{x}^\star\|^2 - \|\boldsymbol{x}_{t+1} - \boldsymbol{x}^\star\|^2 + 2\eta_t h_t(\boldsymbol{x}_t) h_t(\boldsymbol{x}^\star)$$

$$\leq \|\boldsymbol{x}_t - \boldsymbol{x}^\star\|^2 - \|\boldsymbol{x}_{t+1} - \boldsymbol{x}^\star\|^2 + 2\gamma h_t(\boldsymbol{x}^\star) .$$

Taking expectations and using the quadratic growth assumption on $H$, we have

$$\frac{\mu}{2}\min\left(\frac{1}{2L}, \gamma\right) \mathbb{E}\left[\|\boldsymbol{x}_t - \boldsymbol{x}^\star\|^2\right] \leq \min\left(\frac{1}{2L}, \gamma\right) \mathbb{E}[H(\boldsymbol{x}_t) - H(\boldsymbol{x}^\star)]$$

$$\leq \mathbb{E}\left[\|\boldsymbol{x}_t - \boldsymbol{x}^\star\|^2\right] - \mathbb{E}\left[\|\boldsymbol{x}_{t+1} - \boldsymbol{x}^\star\|^2\right] + \left(2\gamma - \min\left(\frac{1}{2L}, \gamma\right)\right) H(\boldsymbol{x}^\star) .$$

---

[5]Using the inequality $\sqrt{z + y} \leq \sqrt{z} + \sqrt{y} \quad \forall z, y \geq 0$.

Hence, we obtain

$$\mathbb{E}\left[\|\boldsymbol{x}_{t+1} - \boldsymbol{x}^\star\|^2\right] \leq (1-a)\mathbb{E}\left[\|\boldsymbol{x}_t - \boldsymbol{x}^\star\|^2\right] + b,$$

where $a = \frac{\mu}{2}\min\left(\frac{1}{2L}, \gamma\right)$ and $b = \left(2\gamma - \min\left(\frac{1}{2L}, \gamma\right)\right)H(\boldsymbol{x}^\star)$. Note we have $0 \leq a \leq \mu/4L \leq 1$. From this inequality, it is immediate to obtain

$$\mathbb{E}\left[\|\boldsymbol{x}_{T+1} - \boldsymbol{x}^\star\|^2\right] \leq \mathbb{E}\left[\|\boldsymbol{x}_1 - \boldsymbol{x}^\star\|^2\right] a^{T+1} + b\frac{1 - a^{T+1}}{1 - a} \,. \qquad \square$$

# D   Additional Convergence Result for Generalized Polyak Stepsize

**Corollary 1.** *Let* $f : \mathbb{R}^d \times \mathcal{S}$ *be convex and L-self-bounded. Define* $\boldsymbol{x}^\star = \operatorname{argmin}_{\boldsymbol{x}}\left(\mathbb{E}_{\boldsymbol{\xi}\sim D}[f(\boldsymbol{x},\boldsymbol{\xi})] := F(\boldsymbol{x})\right)$. *Let* $h(\boldsymbol{x},\boldsymbol{\xi}) = (f(\boldsymbol{x},\boldsymbol{\xi}) - f(\boldsymbol{x}^\star,\boldsymbol{\xi}))_+$. *Then, running Algorithm 1 with* $\gamma = \infty$, *we have*

$$\mathbb{E}[F(\bar{\boldsymbol{x}}_T)] - F(\boldsymbol{x}^\star) \leq \frac{2L\|\boldsymbol{x}_1 - \boldsymbol{x}^\star\|^2}{T} + \frac{\mathbb{E}[\sqrt{2L(F(\boldsymbol{x}^\star) - \mathbb{E}[\inf_{\boldsymbol{x}} f(\boldsymbol{x},\boldsymbol{\xi})])}]\|\boldsymbol{x}_1 - \boldsymbol{x}^\star\|}{\sqrt{T}},$$

*where* $\bar{\boldsymbol{x}}_T = \frac{1}{T}\sum_{t=1}^{T}\boldsymbol{x}_t$ *or* $\bar{\boldsymbol{x}}_T = \operatorname{argmin}_{\boldsymbol{x}\in\{\boldsymbol{x}_1,\ldots,\boldsymbol{x}_T\}} F(\boldsymbol{x})$.

*Proof.* Given the definition of $h$, we have that $H(\boldsymbol{x}^\star) = 0$. Moreover, $h(\boldsymbol{x},\boldsymbol{\xi}) \geq f(\boldsymbol{x},\boldsymbol{\xi}) - f(\boldsymbol{x}^\star,\boldsymbol{\xi})$, hence $\mathbb{E}[H(\boldsymbol{x}_t)] \geq \mathbb{E}[F(\boldsymbol{x}_t)] - F(\boldsymbol{x}^\star)$ for any $t$.

We have that

$$\partial h(\boldsymbol{x},\boldsymbol{\xi}) = \begin{cases} \partial f(\boldsymbol{x},\boldsymbol{\xi}), & \text{if } f(\boldsymbol{x},\boldsymbol{\xi}) > f(\boldsymbol{x}^\star,\boldsymbol{\xi}) \\ \{0\}, & \text{if } f(\boldsymbol{x},\boldsymbol{\xi}) < f(\boldsymbol{x}^\star,\boldsymbol{\xi}) \\ \{\alpha\boldsymbol{g} : \alpha \in [0,1], \boldsymbol{g} \in \partial f(\boldsymbol{x},\boldsymbol{\xi})\}, & \text{if } f(\boldsymbol{x},\boldsymbol{\xi}) = f(\boldsymbol{x}^\star,\boldsymbol{\xi}) \end{cases} .$$

Hence, for all $\boldsymbol{g}_t \in \partial h(\boldsymbol{x}_t,\boldsymbol{\xi}_t)$ we have

$$\begin{aligned}
\|\boldsymbol{g}_t\|^2 &\leq 2L(f(\boldsymbol{x},\boldsymbol{\xi}_t) - \inf_{\boldsymbol{x}} f(\boldsymbol{x},\boldsymbol{\xi}_t)) \\
&= 2L(f(\boldsymbol{x},\boldsymbol{\xi}_t) - f(\boldsymbol{x}^\star,\boldsymbol{\xi}_t) + f(\boldsymbol{x}^\star,\boldsymbol{\xi}_t) - \inf_{\boldsymbol{x}} f(\boldsymbol{x},\boldsymbol{\xi}_t)) \\
&\leq 2L(h(\boldsymbol{x},\boldsymbol{\xi}_t) + f(\boldsymbol{x}^\star,\boldsymbol{\xi}_t) - \inf_{\boldsymbol{x}} f(\boldsymbol{x},\boldsymbol{\xi}_t)) \,.
\end{aligned}$$

So, using this inequality in Lemma 3 gives

$$\sum_{t=1}^{T} \frac{h_t^2(\boldsymbol{x}_t)}{4L(h_t(\boldsymbol{x}_t) + f_t(\boldsymbol{x}^\star) - f_t^\star)} \leq \sum_{t=1}^{T} \frac{h_t^2(\boldsymbol{x}_t)}{2\|\boldsymbol{g}_t\|^2} = \sum_{t=1}^{T} \eta_t \frac{1}{2} h_t^2(\boldsymbol{x}_t) \leq \frac{1}{2}\|\boldsymbol{x}_1 - \boldsymbol{x}^\star\|^2 \,. \qquad (8)$$

Now, from Cauchy–Schwarz inequality, for any non-negative random variable $Y$ and random variable $X$, we have $\mathbb{E}[X^2/Y] \geq (\mathbb{E}[X])^2/\mathbb{E}[Y]$. Denote by $f_t^\star = \inf_{\boldsymbol{x}} f(\boldsymbol{x},\boldsymbol{\xi}_t)$. Given that $f_t(\boldsymbol{x}^\star) - f_t^\star \geq 0$, if $f_t(\boldsymbol{x}^\star) - f_t^\star = 0$ with probability 1, i.e., $F(\boldsymbol{x}^\star) - \mathbb{E}[\inf_{\boldsymbol{x}} f(\boldsymbol{x},\boldsymbol{\xi})] = 0$, then the expectation of the l.h.s. of the previous inequality is $\mathbb{E}[h_t(\boldsymbol{x}_t)]$. Otherwise, if we assume $F(\boldsymbol{x}^\star) - \mathbb{E}[\inf_{\boldsymbol{x}} f(\boldsymbol{x},\boldsymbol{\xi})] > 0$, we have

$$\mathbb{E}\left[\frac{h_t^2(\boldsymbol{x}_t)}{4L(h_t(\boldsymbol{x}_t) + f_t(\boldsymbol{x}^\star) - f_t^\star)}\right] \geq \frac{(\mathbb{E}[h_t(\boldsymbol{x}_t)])^2}{4L(\mathbb{E}[h_t(\boldsymbol{x}_t)] + F(\boldsymbol{x}^\star) - \mathbb{E}[\inf_{\boldsymbol{x}} f(\boldsymbol{x},\boldsymbol{\xi})])} \,.$$

Hence, in all cases we have the last expression is a lower bound to the l.h.s. of (8). We now can proceed as in the proof of the Lipschitz case in Theorem 3, to have the stated bound. $\qquad \square$

We now extend Theorem 3 to Hölder-self-bounded functions.

**Definition 6.** *We say that* $f$ *is* $(L_\nu, \nu)$ *Hölder-self-bounded if there exits* $\nu \in [0,1]$ *and* $L_\nu$ *such That*

$$\|\boldsymbol{g}\|^2 \leq \left(1 + \frac{1}{\nu}\right)^{\frac{2\nu}{1+\nu}} L_\nu^{\frac{2}{1+\nu}} (f(\boldsymbol{x}) - f(\boldsymbol{x}^\star))^{\frac{2\nu}{1+\nu}}, \ \forall \boldsymbol{g} \in \partial f(\boldsymbol{x}) \,.$$

This definition is weaker than both Lipschitz and smoothness and it is easy to see that $L$-smooth functions satisfies this condition with $\nu = 1$ and $L_1 = L$.

The following theorem generalizes both the Lipschitz and the smooth case, recovering both bounds up to constant factors.

**Theorem 7.** *Let $h : \mathbb{R}^d \times \mathcal{S} \to \mathbb{R}_{\geq 0}$ be convex. Denote by $H(\boldsymbol{x}) = \mathbb{E}_{\boldsymbol{\xi} \sim D}[h(\boldsymbol{x}, \boldsymbol{\xi})]$. Assume that $h(\cdot, \boldsymbol{\xi}_t)$ is $(L_\nu, \nu)$-Hölder-self bounded. Then, setting $\eta_t = \min\left(\frac{1}{\|\boldsymbol{g}_t\|^2}, \frac{\gamma}{h(\boldsymbol{x}_t, \boldsymbol{\xi}_t)}\right)$ in Algorithm 1, we have*

$$\frac{1}{T}\sum_{t=1}^{T}\mathbb{E}[H(\boldsymbol{x}_t)] \leq Q\left(\frac{\|\boldsymbol{x}_1 - \boldsymbol{x}^\star\|^2}{T\gamma} + 2H(\boldsymbol{x}^\star)\right),$$

*where $Q(y) = 2y + L_\nu(2\gamma y)^{\frac{1+\nu}{2}}\left(1 + \frac{1}{\nu}\right)^\nu$.*

*Proof.* For simplicity, denote by $h_t(\boldsymbol{x}) = h(\boldsymbol{x}, \xi_t)$.

From the Lemma 3, we have

$$\sum_{t=1}^{T}\eta_t \frac{1}{2}h_t^2(\boldsymbol{x}_t) \leq \frac{1}{2}\|\boldsymbol{x}_1 - \boldsymbol{x}^\star\|^2 + \sum_{t=1}^{T}\frac{\eta_t}{2}\left(\eta_t - \frac{1}{\|\boldsymbol{g}_t\|^2}\right)\|\boldsymbol{g}_t\|^2 h_t^2(\boldsymbol{x}_t) + \sum_{t=1}^{T}\eta_t h_t(\boldsymbol{x}_t)h_t(\boldsymbol{x}^\star).$$

For the last term in the r.h.s., we have

$$\eta_t h_t(\boldsymbol{x}_t)h_t(\boldsymbol{x}^\star) = \min\left(\frac{1}{\|\boldsymbol{g}_t\|^2}, \frac{\gamma}{h_t(\boldsymbol{x}_t)}\right)h_t(\boldsymbol{x}_t)h_t(\boldsymbol{x}^\star) \leq \gamma h_t(\boldsymbol{x}^\star).$$

Observe that if $h_t$ is $(L_\nu, \nu)$-Hölder-self-bounded then

$$\|g_t\|^2 \leq \left(1 + \frac{1}{\nu}\right)^{\frac{2\nu}{1+\nu}}L_\nu^{\frac{2}{1+\nu}}\left(h_t(\boldsymbol{x}) - \inf_{\boldsymbol{x}}h_t(\boldsymbol{x})\right)^{\frac{2\nu}{1+\nu}} \leq K_\nu h_t(\boldsymbol{x})^{\frac{2\nu}{1+\nu}},$$

where $K_\nu = \left(1 + \frac{1}{\nu}\right)^{\frac{2\nu}{1+\nu}}L_\nu^{\frac{2}{1+\nu}}$. Therefore, we have

$$\min\left(\frac{h_t^{\frac{1-\nu}{1+\nu}}(\boldsymbol{x}_t)}{K_\nu}, \gamma\right)h_t(\boldsymbol{x}_t) \leq \min\left(\frac{h_t(\boldsymbol{x}_t)}{\|\boldsymbol{g}_t\|^2}, \gamma\right)h_t(\boldsymbol{x}_t) = \eta_t h_t^2(\boldsymbol{x}_t).$$

As before, we lower bound the minimum with the convex function $B(x) = \frac{x^{\frac{2}{1+\nu}}}{x^{\frac{1-\nu}{1+\nu}} + \gamma K_\nu}$:

$$\min\left(\frac{h_t^{\frac{1-\nu}{1+\nu}}(\boldsymbol{x}_t)}{K_\nu}, \gamma\right)h_t(\boldsymbol{x}_t) \geq \frac{\gamma h_t^{\frac{2}{1+\nu}}(\boldsymbol{x}_t)}{h_t^{\frac{1-\nu}{1+\nu}}(\boldsymbol{x}_t) + \gamma K_\nu} = \gamma B(h_t(\boldsymbol{x}_t)).$$

As before, this allows us to use Jensen's inequality:

$$B\left(\frac{1}{T}\sum_{t=1}^{T}h_t(\boldsymbol{x}_t)\right) \leq \frac{1}{T}\sum_{t=1}^{T}B(h_t(\boldsymbol{x}_t)) \leq \frac{\|\boldsymbol{x}_1 - \boldsymbol{x}^\star\|^2}{\gamma T} + \frac{2}{T}\sum_{t=1}^{T}h_t(\boldsymbol{x}^\star).$$

For simplicity of calculations, we now lower bound $B(x)$,

$$C(x) := 0.5\min\left(x, \frac{x^{\frac{2}{1+\nu}}}{\gamma K_\nu}\right) = \frac{x^{\frac{2}{1+\nu}}}{2\max\{x^{\frac{1-\nu}{1+\nu}}, \gamma K_\nu\}} \leq B(x).$$

Note that $C(x)$ is invertible and its inverse is

$$C^{-1}(y) = \begin{cases} 2y, & \text{if } y \geq (\gamma K_\nu)^{\frac{1+\nu}{1-\nu}} \\ (2\gamma K_\nu y)^{\frac{1+\nu}{2}}, & \text{if } y < (\gamma K_\nu)^{\frac{1+\nu}{1-\nu}} \end{cases}$$

$$\leq 2y + (2\gamma K_\nu y)^{\frac{1+\nu}{2}}$$

$$= 2y + L_\nu(2\gamma y)^{\frac{1+\nu}{2}}\left(1 + \frac{1}{\nu}\right)^\nu.$$

Taking expectations and using Jensen's inequality gives the stated bound. $\qquad\square$

# E  Proofs for Section 6

**Lemma 4.** *Let $f : \mathbb{R}^n \to \mathbb{R}^+$ where $f^\star = \inf_{\boldsymbol{x}} f(\boldsymbol{x})$. Then for any $c \geq 0$ the following are equivalent:*

- $f(\boldsymbol{x}) - f^\star < cf^\star$,
- $f(\boldsymbol{x}) - f^\star < \frac{c}{c+1} f(\boldsymbol{x})$.

*Proof.*

$$f(\boldsymbol{x}) - f^\star < cf^\star \Leftrightarrow f(\boldsymbol{x}) < (c+1)f^\star$$

$$\Leftrightarrow \frac{f(\boldsymbol{x})}{(c+1)} < f^\star$$

$$\Leftrightarrow f(\boldsymbol{x}) - f^\star < \left(1 - \frac{1}{c+1}\right) f(\boldsymbol{x}) = \frac{c}{c+1} f(\boldsymbol{x}) . \qquad \square$$

**Proposition 1.** *Suppose $h$ is convex, strictly positive, $L$-self-bounded, and satisfies the quadratic growth condition $h(\boldsymbol{x}) - h^\star \geq \frac{\mu}{2}\|\boldsymbol{x} - \boldsymbol{x}^\star\|^2$, where $\boldsymbol{x}^\star = \arg\min_{\boldsymbol{x}} h(\boldsymbol{x})$ is the only fixed point of $T$ (6). Then for any point $\boldsymbol{x} \in S = \{\boldsymbol{y} : h(\boldsymbol{y}) - h^\star < h^\star \frac{\mu}{8L-\mu}\}$ we have*

$$\|T(\boldsymbol{x}) - \boldsymbol{x}^\star\| > \|\boldsymbol{x} - \boldsymbol{x}^\star\| .$$

*Proof.* Let $\boldsymbol{x}_t$ be in $S$ then by definition of $T$ we have

$$\frac{1}{2}\|\boldsymbol{x}_{t+1} - \boldsymbol{x}^\star\|^2 = \frac{1}{2}\|\boldsymbol{x}_t - \boldsymbol{x}^\star\|^2 - \eta_t \langle \boldsymbol{g}_t, \boldsymbol{x}_t - \boldsymbol{x}^\star \rangle + \frac{\eta_t^2}{2}\|\boldsymbol{g}_t\|^2$$

$$\geq \frac{1}{2}\|\boldsymbol{x}_t - \boldsymbol{x}^\star\|^2 - \eta_t \|\boldsymbol{g}_t\|\|\boldsymbol{x}_t - \boldsymbol{x}^\star\| + \frac{\eta_t^2}{2}\|\boldsymbol{g}_t\|^2$$

$$= \frac{1}{2}\|\boldsymbol{x}_t - \boldsymbol{x}^\star\|^2 - \frac{h(\boldsymbol{x}_t)}{\|\boldsymbol{g}_t\|}\|\boldsymbol{x}_t - \boldsymbol{x}^\star\| + \frac{h(\boldsymbol{x}_t)^2}{2\|\boldsymbol{g}_t\|^2}$$

$$= \frac{1}{2}\|\boldsymbol{x}_t - \boldsymbol{x}^\star\|^2 + \frac{h(\boldsymbol{x}_t)}{\|\boldsymbol{g}_t\|}\left[\frac{h(\boldsymbol{x}_t)}{2\|\boldsymbol{g}_t\|} - \|\boldsymbol{x}_t - \boldsymbol{x}^\star\|\right] .$$

If $h(\boldsymbol{x}_t) - h^\star < h^\star \frac{\mu}{8L-\mu}$ then we have by Lemma 4

$$h(\boldsymbol{x}_t) > \left(\frac{\mu/(8L-\mu) + 1}{\mu/(8L-\mu)}\right)(h(\boldsymbol{x}_t) - h^\star) = \frac{8L}{\mu}(h(\boldsymbol{x}_t) - h^\star) .$$

Consequently,

$$\frac{h(\boldsymbol{x}_t)}{2\|\boldsymbol{g}_t\|} > \frac{4L}{\mu}\frac{(h(\boldsymbol{x}_t) - h(\boldsymbol{x}^\star))}{\|\boldsymbol{g}_t\|} \geq 2L\frac{\|\boldsymbol{x}_t - \boldsymbol{x}^\star\|^2}{\|\boldsymbol{g}_t\|} \geq \|\boldsymbol{x}_t - \boldsymbol{x}^\star\| .$$

Where the last inequality follows from $h$ being self-bounded and convex,

$$\frac{1}{2L}\|\boldsymbol{g}_t\|^2 \leq h(\boldsymbol{x}_t) - h^\star \leq \langle \boldsymbol{g}_t, \boldsymbol{x}_t - \boldsymbol{x}^\star \rangle \leq \|\boldsymbol{g}_t\|\|\boldsymbol{x}_t - \boldsymbol{x}^\star\| . \qquad \square$$

**Proposition 6.** *Suppose $h$ is convex, strictly positive, $L$-Lipschitz, and has a $\mu$-sharp minimum $h(\boldsymbol{x}) - h^\star \geq \mu\|\boldsymbol{x} - \boldsymbol{x}^\star\|$, where $\boldsymbol{x}^\star = \arg\min_{\boldsymbol{x}} h(\boldsymbol{x})$ is the only fixed point of $T$ (6). Then for any point $\boldsymbol{x} \in S = \{\boldsymbol{y} : \boldsymbol{y} \neq \boldsymbol{x}^\star, h(\boldsymbol{y}) - h^\star < h^\star \frac{\mu}{2L-\mu}\}$ we have*

$$\|T(\boldsymbol{x}) - \boldsymbol{x}^\star\| > \|\boldsymbol{x} - \boldsymbol{x}^\star\| .$$

*Proof.* Let $\boldsymbol{x}_t \in S$ then by following similar steps to Lemma 1 we have

$$\frac{1}{2}\|\boldsymbol{x}_{t+1} - \boldsymbol{x}^\star\|^2 \geq \frac{1}{2}\|\boldsymbol{x}_t - \boldsymbol{x}^\star\|^2 + \frac{h(\boldsymbol{x}_t)}{\|\boldsymbol{g}_t\|}\left[\frac{h(\boldsymbol{x}_t)}{2\|\boldsymbol{g}_t\|} - \|\boldsymbol{x}_t - \boldsymbol{x}^\star\|\right] .$$

By Lemma 4, we have

$$h(\boldsymbol{x}_t) - h^\star < h^\star \frac{\mu}{2L - \mu}$$

$$\Leftrightarrow h(\boldsymbol{x}_t) > \left( \frac{\mu/(2L-\mu) + 1}{\mu/(2L-\mu)} \right) (h(\boldsymbol{x}_t) - h^\star) = \frac{2L}{\mu} (h(\boldsymbol{x}_t) - h^\star) \,.$$

Therefore, by sharpness and Lipschitz property of $h$, we have

$$\frac{h(\boldsymbol{x}_t)}{2\|\boldsymbol{g}_t\|} > \frac{L(h(\boldsymbol{x}_t) - h^\star)}{\mu\|\boldsymbol{g}_t\|} \geq \frac{L\|\boldsymbol{x} - \boldsymbol{x}^\star\|}{\|\boldsymbol{g}_t\|} \geq \|\boldsymbol{x}_t - \boldsymbol{x}^\star\| \,. \qquad \square$$

**Proposition 2** (Cycling and failure to converge). *There exists a strictly positive smooth and strongly convex function $h$, and initial point $\boldsymbol{x}_1$ such that iterates from update* (6) *cycle and satisfy the inequality $h(\frac{1}{t}\sum_{i=1}^{t} \boldsymbol{x}_i) - h^\star \geq \delta > 0$ for all $t$.*

*Proof.* The proof is constructive: consider $h : \mathbb{R} \to \mathbb{R}, h(x) = x^2 + 1$, so $h^\star = 1$. Observe that the update is

$$x_{t+1} = x_t - \frac{x_t^2 + 1}{2x_t} = \frac{x_t^2 - 1}{2x_t} \,.$$

Now, we want to choose $x_1$ so that we oscillate between 3 possible values.

Set $x_1 = \cot\theta$ where $\theta$ has to be determined. The update becomes

$$x_2 = \frac{x_1^2 - 1}{2x_1} = \frac{\cot^2\theta - 1}{2\cot\theta} = \cot(2\theta),$$

where in the last equality we used the identity for $\cot$. Hence, we have $x_t = \cot(2^t\theta)$. Given that we want to oscillate between 3 values, we want $x_{t+3} = x_t$, that is, $\cot(2^{t+3}\theta) = \cot(2^t\theta)$. We can achieve it if we select $\theta = \pi/7$. Indeed, we have

$$x_1 = \cot(\pi/7)$$
$$x_2 = \cot(2\pi/7)$$
$$x_3 = \cot(4\pi/7)$$
$$x_4 = \cot(8\pi/7) = \cot(\pi + \pi/7) = \cot(\pi/7) = x_1 \,.$$

Finally, one can verify numerically that $f(\frac{1}{t}\sum_{i=1}^{t} x_t) - f^\star > 0.77$. $\qquad \square$

**Proposition 8.** *There exists subregions within the unstable regions in Propositions 1 and 6 where the stepsizes are upper bounded.*

*Proof.* By convexity of $h$ we have $h - h^\star \leq \langle \boldsymbol{g}_t, \boldsymbol{x}_t - \boldsymbol{x}^\star \rangle \leq \|\boldsymbol{g}_t\|\|\boldsymbol{x}_t - \boldsymbol{x}^\star\|$, so $\|\boldsymbol{g}_t\| \geq \frac{h(\boldsymbol{x})-h^\star}{\|\boldsymbol{x}_t - \boldsymbol{x}^\star\|}$. By assumption in Lemmas 1 and 6 the unstable region is $S = \{\boldsymbol{x} : h(\boldsymbol{x}) - h^\star < ch^\star\}$ where $c$ depends on the properties of $h$. Therefore, for $\boldsymbol{x} \in S$ and denoting $\boldsymbol{g} \in \partial h(\boldsymbol{x})$ as any subgradient at $\boldsymbol{x}$, we have

$$\frac{h(\boldsymbol{x})}{\|\boldsymbol{g}\|^2} \leq \frac{h(\boldsymbol{x})\|\boldsymbol{x} - \boldsymbol{x}^\star\|^2}{(h(\boldsymbol{x}) - h^\star)^2} < \frac{(c+1)h^\star\|\boldsymbol{x} - \boldsymbol{x}^\star\|^2}{(h(\boldsymbol{x}) - h^\star)^2} \,.$$

If $h$ has a sharp minimum, $h(\boldsymbol{x}) - h^\star \geq \mu\|\boldsymbol{x} - \boldsymbol{x}^\star\|$, then we have $\frac{h(\boldsymbol{x})}{\|\boldsymbol{g}_t\|^2} < \frac{c+1}{\mu^2}h^\star$. Therefore, the stepsizes are always bounded within S.

Now consider the subregion $S_k = \{\boldsymbol{x} : h(\boldsymbol{x}) - h^\star < \frac{c}{k}h^\star\}$ for some $k > 1$. Consider $\boldsymbol{x} \in S \setminus S_k$, that is $(1 + \frac{c}{k})h^* \leq h(\boldsymbol{x}) \leq (1 + c)h^*$. If $h$ statisfies the quadratic growth condition $h(\boldsymbol{x}) - h^\star \geq \frac{\mu}{2}\|\boldsymbol{x} - \boldsymbol{x}^\star\|^2$ then

$$\frac{h(\boldsymbol{x})}{\|\boldsymbol{g}\|^2} < \frac{(c+1)h^\star\|\boldsymbol{x} - \boldsymbol{x}^\star\|^2}{(h(\boldsymbol{x}) - h^\star)^2} \leq \frac{2(c+1)h^\star}{\mu(h(\boldsymbol{x}) - h^\star)} \leq \frac{2k(c+1)}{\mu c} \,. \qquad (9)$$

Where the last inequality follows since $\boldsymbol{x} \notin S_k$. Therefore, stepsize is bounded within $S \setminus S_k$, and grows as we increase $k$. $\qquad \square$

**Definition 7** (Lusin $(N^{-1})$ condition). *Let $T : \mathbb{R}^d \to \mathbb{R}^k$. We define $T^{-1}$ over a set $S \subseteq \mathbb{R}^k$ as*

$$T^{-1}(S) = \{\boldsymbol{x} : T(\boldsymbol{x}) \in S\}.$$

*We say that $T$ satisfies $(N^{-1})$ condition if for every set $E$ of measure zero we have that $T^{-1}(E)$ also has measure zero.*

**Lemma 5.** *Let $T : \mathbb{R}^n \to \mathbb{R}^n$ with a unique fixed point $\boldsymbol{x}^\star$. If $\boldsymbol{x}^\star$ is unstable, that is, there exists $\delta$ such that $\boldsymbol{x} \neq \boldsymbol{x}^\star$ and $\|\boldsymbol{x} - \boldsymbol{x}^\star\| \leq \delta$, then $\|T(\boldsymbol{x}) - \boldsymbol{x}^\star\| > \|\boldsymbol{x} - \boldsymbol{x}^\star\|$. Define $T^{-1}$ over a set $S \subseteq \mathbb{R}^n$ as $T^{-1}(S) = \{\boldsymbol{x} : T(\boldsymbol{x}) \in S\}$. If $T^{-k}(\{\boldsymbol{x}^\star\})$ is of measure zero for any $k$, then*

$$P\left(\lim_{t \to \infty} \boldsymbol{x}_t = \boldsymbol{x}^\star\right) = 0 \,.$$

*In other words, the set of initializations that can converge to the fixed point has measure zero.*

*Proof.* We divide $\mathbb{R}^n$ into two sets, $S = \bigcup_{k=1}^{\infty} T^{-k}(\{\boldsymbol{x}^\star\})$, and its compliment $S^c$. $S$ represents the points that can exactly reach the unique minimizer $\boldsymbol{x}^\star$. If $T^k(\{\boldsymbol{x}^\star\})$ is a null set for every $k$ then so is $S$ since the countable union of null sets is a null set.

Now we show that for all initializations in $\boldsymbol{x}_1 \in S^c$, $\boldsymbol{x}_1$ cannot converge to $\boldsymbol{x}^\star$. Suppose the contrary, $\lim_{t \to \infty} \boldsymbol{x}_t = \boldsymbol{x}^\star$. Let $B = \{\boldsymbol{y} : \boldsymbol{y} \neq \boldsymbol{x}^\star, \|\boldsymbol{x} - \boldsymbol{x}^\star\| \leq \delta\}$. Since $\boldsymbol{x}_t \to \boldsymbol{x}^\star$ there exists a step $n$ where $\{\boldsymbol{x}_t\}_{t \geq n} \subseteq B$. Similarly, there exists $n' \geq n$ where $\|\boldsymbol{x}_t - \boldsymbol{x}^\star\| \leq \|\boldsymbol{x}_n - \boldsymbol{x}^\star\|$ for all $t \geq n'$. This is a contradiction as we have that $\|\boldsymbol{x}_n - x^\star\| < \|\boldsymbol{x}_{n+1} - \boldsymbol{x}^\star\| < \cdots < \|\boldsymbol{x}_{n'} - \boldsymbol{x}^\star\|$. Therefore, the initializations that allow for $\boldsymbol{x}_t \to \boldsymbol{x}^\star$ coincide exactly with the null set $S$. $\qquad\square$

