# OpenReview forum: "New Perspectives on the Polyak Stepsize: Surrogate Functions and Negative Results"
_NeurIPS.cc/2025/Conference — NeurIPS 2025 poster_

### Official Review · Reviewer_AKhF · 2025-06-25

**Clarity:** 2
**Significance:** 2
**Originality:** 2
**Rating:** 3
**Confidence:** 4

**Summary:**

This paper sheds light on the Polyak step size by relating it to gradient descent with curvature-inspired step sizes on a surrogate function. This connection is used in order to provide negative results for Polyak step sizes, for example in the stochastic setting without interpolation, or when the optimal value is underestimated.

**Questions:**

- From the definition of the subgradients, it seems that only convex $f$ are considered, but the introduction remains unclear on this.
- I think the statement of Lemma 1 is unprecise: first, the LSUC assumption is actually needed for $\phi$ and not $f$ as stated. Second, the Lemma also uses the fact that $\phi(x^*)=0$, but this assumption is not mentioned. Finally, the proof uses the notation $g_t$ and $g_{x_t}$ which in my understanding denote the same vector.
- line 153: I get to the linear rate $1-\frac{2s^2}{G^2}$ which somehow could be negative if $s^2 = G^2$. What am I missing?
- line 179: "The last two lemmas tells us that the properties of the surrogate functions breaks if $\psi(x*) \neq 0$." Which properties are meant here exactly? Can you elaborate how this can be seen from the Lemmas?
- Theorem 3: Do you need to assume that $h$ is jointly convex in the two arguments? Convexity in $x$ should be sufficient.
- I am not sure if the second part of Remark 1 is correct: consider $\xi$ having two possible values and $f_1(x)=|x-1|$ and $f_2(x)=|x+1|$. Then $a=0$ satisifies the condition, but if $x^* $ is the minimizer of $1/2 (f_1 + f_2)$, then for one of the two $\xi$ we have $h(x^*,\xi) > 0$.
- line 225: This is not correct; the paper by Loizou et al. has results for the nonsmooth setting in the Appendix C.1. (However they only have results where the SPS step size is not capped at $\gamma$).
- line 266: I could not find a Proposition 5 neither in the main text nor in the appendix

Minor:

- line 29: "adaptive to" missing space
- line 60: Incomplete sentence
- line 83: "they will BE denoted"
- line 89: "Note that IF"
- line 97: "chain rule of subgradients". Maybe better to say "chain rule of convex subgradients" as there are notions of subgradients that do not follow a chain rule.
- line 140: "the last iterate will converge". I think it is more accurate to say "the function values converge to zero" (here the optimal value of $\phi$ is zero, see above)

**Ethical Concerns:**

["NO or VERY MINOR ethics concerns only"]

**Final Justification:**

In the discussion period some smaller points of confusion have been clarified. However, I remain the borderline score as the main weaknesses mentioned in the initial review remain open: it is far from obvious how to use the framework of this paper to design better practical Polyak stepsize variants. Can the surrogate function viewpoint explain the Polyak stepsize dynamics that occurr for the problems of interest (i.e. ML training problems)?

**Paper Formatting Concerns:**

Please concatenate the appendix with the main text PDF.

**Quality:**

3

**Strengths And Weaknesses:**

**Strengths:**

- The results of Section 6 are interesting, showing that the negative results are not an artefact of the analysis. They shed a lower-bound-type insight on the empirical observations that Polyak step sizes are less powerful for non-interpolating problems.
- The connection to the surrogate function could explain the adaptivity of the Polyak step size (see also Weaknesses below).

**Weaknesses:**

- Some of the formal statements of the paper need to be formulated more precisely in order to avoid confusion (see below for details).
- From the connection to the surrogate function, some points remains open: why is it seemingly beneficial to adapt to the curvature of the surrogate function? Does it explain the dynamics of the Polyak stepsize in practice (for example, it has been observed that the Polyak step size often warms up and then decays for ML problems)?
- On the conclusion:
	>  We believe this framework has the promise to design new variants, by simply designing surrogate functions with the required properties.

This sounds promising, but in fact all surrogate functions discussed in the paper have been previously proposed (which is also mentioned in the paper). No attempts are made to explore new surrogate functions beyond the existing ones. The paper would benefit a lot in my opinion from such explorations (which could be tested by experiments); currently, it remains unclear whether the connection to surrogate function is useful beyond proving negative results.

- The paper makes a few considerations which could be analyzed beyond the theory: for example, it mentions that non-convex problems are harder to analyze theoretically. However, it could be investigated through experiments if the presented limitations of Polyak step sizes can be observed also on non-convex problems (e.g. cycling behaviour due to underestimation in the deterministic case). Another interesting direction would be the proposed rule of $a= \sup_\xi \inf_x f(x,\xi)$. How does this rule perform in a practical setup (constructed such that $a$ can be computed)? Can we estimate $a$ somehow on the fly? In short: I am not proposing to add experiments for the sake of having experiments, but because they could clarify and complete the story of the paper.

- If I understand correctly, the conclusion says that the neighborhood of convergence is unaviodable without interpolation. This is not accurate: the paper [17] shows that the convergence rates are achieved in the smooth and non-smooth case even without interpolation, but assuming that we know $f(x^*,\xi)$ for all $\xi$ (which of course is a very strong assumption, but it is not the same as unavoidable).

---

> ### Author Rebuttal · Authors · 2025-07-30
>
> - it remains unclear whether the connection to surrogate function is useful beyond proving negative results.
>
> This is exactly our main point: In light of our negative results, proposing yet another stochastic variant of the Polyak stepsize in the case where the optimum value of $f$ is unknown is not a promising direction. Judging from the number of papers that ignore this issue, this seems a very much needed point for this community. So, our negative results are not a minor point, but very much an essential part of our contribution. In turn, as recognized by the reviewer, these results are a consequence of our surrogate view.
>
> - the conclusion says that the neighborhood of convergence is unavoidable without interpolation
>
> This is not what we meant. We wrote “We demonstrate that this neighbourhood is not avoidable
> and a fundamental issue beyond interpolation causing instability”. So, what we mean is that the neighbourhood of convergence is not due to the lack of interpolation, as commonly believed, but instead on the fact that the minimum of the surrogate loss is not zero. This condition is formally expressed in Eq. (5) in Section 6.
> We believe that this is a very important point that is possibly misunderstood by many people in this field.
>
> -  it seems that only convex function are considered
>
> Correct, we will make it more clear.
>
> - Typos in Lemma 1
>
> You are correct, we will fix them.
>
> - Rate for sharp functions
>
> As the other reviewers also found, there is a typo, a missing “1/2" in the first two terms of the inequalities
>
> - Which properties do not hold if $\psi(x^*)\neq 0$?
>
> In this case, the function $\psi$ is only approximately LSUC (Lemma 2) instead of being LSUC. This implies an additional term in Lemma 3. As discussed in Section 6, we show that the consequence of this extra term cannot be avoided.
>
> - Convexity of h in Theorem 3 is only required for x
>
> Correct! We will add it.
>
> - second part of Remark 1
>
> Sorry that is a weird typo, we meant $a = \sup_\xi f(x^*,\xi)$.
>
> - the paper by Loizou et al. has results for the nonsmooth setting in the Appendix C.1
>
> At the end of Section 5 were we discuss the non-smooth results in Loizou et al we are referring to $SPS_{\max}$, as mentioned in the following sentence [225-226], and implied by the context of our Theorem 3 (using stepsize from Algorithm 1). In [1] there is no result for
>  $SPS_{\max}$ in the non-smooth case. In [1] they only have a nonsmooth result for SPS in the easier interpolation case. This is not a minor difference: They claim that “Using the proof techniques from the rest of the paper one can easily obtain convergence for the more general setting”, but this does not seem to be true. The technical reason is that the min function would introduce a non-convexity that would not allow the use of Jensen’s inequality. We show how to go around it, by lower bounding it with a convex function in line 805 in the supplemental material. So, a proof is indeed possible but not with the same tools they use. We will be more precise on this point and make it clear that we are referring to $SPS_{\max}$
>
> - Where is Proposition 5?
>
> Typo, it should be Lemma 5 (as you can see from the hyperlink in the pdf).
>
> - "the last iterate will converge". I think it is more accurate to say "the function values converge to zero" (here the optimal value of is zero)
>
> Thank you for the clarifying question, in this case we do in fact we do have convergence of the last iterate to a point in the solution set. This question was also asked by reviewer qnH4 and we provide a similar response below. We will also include this discussion in the Appendix.
>
> The result follows from a classic argument with Fejér monotone sequences. From Lemma 1 we have that the distance to any solution is decreasing, that is, $\\{\boldsymbol{x}_t\\}$ is a Fejér monotone sequence with respect to the solution set. Since we have that $\phi(\boldsymbol{x}_t)\to 0$ and $\phi$ is continuous then we have that every limit point $\\{\boldsymbol{x}’\\}$ is within the solution set since $\phi(\boldsymbol{x}’=0)$. Therefore we can use the fact that if $\\{\boldsymbol{x}_t\\}_t$ is Fejér monotone with respect to the solution set and the set contains all the limit points of the sequence then the sequence must converge to a point in the solution set (see Theorem 8.16 in [Beck 17]).
>
> [Beck 17] Beck, Amir. First-order methods in optimization. Society for Industrial and Applied Mathematics, 2017.

---

> > ### Comment · Reviewer_AKhF · 2025-08-01
> > **Response to rebuttal**
> >
> > Dear authors,
> >
> > thank you for your rebuttal.
> >
> > --------------------------------
> > Re:
> > >  So, what we mean is that the neighbourhood of convergence is not due to the lack of interpolation, as commonly believed, but instead on the fact that the minimum of the surrogate loss is not zero.
> >
> > In my opinion, this formulation is much clearer. I fully agree with the above conclusion and would encourage to rephrase this specific paragraph in order to avoid possible confusion.
> >
> > --------------------------------
> > Re:
> > > This is exactly our main point: In light of our negative results, proposing yet another stochastic variant of the Polyak stepsize in the case where the optimum value of $f$ is unknown is not a promising direction.
> >
> > At the same time the paper says that:
> >
> > > We believe this framework has the promise to design new variants, by simply designing surrogate functions with the required properties.
> >
> > So, what is left to do is to find surrogate functions with the required properties, but in the case where the optimum value is unknown? This does not seem so simple. Your rebuttal does not comment on my original question whether you experimented with any such new methods that would be inspired by the framework (negative results would be interesting as well).
> >
> > I would also like to point out that some other questions in the review have not been responded to, for example:
> >
> > > From the connection to the surrogate function, some points remains open: why is it seemingly beneficial to adapt to the curvature of the surrogate function? [...]
> >
> > I am happy to discuss those further in case the authors can shed any light on it.

---

> > > ### Author Response · Authors · 2025-08-01
> > > **Response to reviewer**
> > >
> > > Thank you for your prompt response, below we answer your two questions above.
> > >
> > > > So, what is left to do is to find surrogate functions with the required properties, but in the case where the optimum value is unknown? This does not seem so simple. Your rebuttal does not comment on my original question whether you experimented with any such new methods that would be inspired by the framework (negative results would be interesting as well).
> > >
> > > We agree with the Reviewer. Indeed, in the stochastic case we find it very difficult to think of suitable surrogate functions if we do not know the optimal value or quantities related to it like in $SPS_+$. However, we do not exclude the possibility to work with weaker proxies of the optimum value. One could also extend our framework by using a sequence of surrogate functions that, for example, improve the estimates of the optimum value over time. However, regretfully, we did not try to work much in this direction.
> > > Instead, as we answered to Reviewer xpPF, we believe that the surrogate view also gives orthogonal directions of research that would be very difficult otherwise, as for example the possibility to design an accelerated variant of the Polyak stepsize. We are currently experimenting in this direction, which seems more promising to us.
> > >
> > >
> > >
> > > > From the connection to the surrogate function, some points remains open: why is it seemingly beneficial to adapt to the curvature of the surrogate function? Does it explain the dynamics of the Polyak stepsize in practice (for example, it has been observed that the Polyak step size often warms up and then decays for ML problems)?
> > >
> > > In regards to the first part, why adapt to local curvature at all? From the math, one can see the advantage of adapting to local curvature. However, it might be useful to have some intuition as well. In the smooth case, one can show a maximum worst-case descent per step by using a stepsize of 1/(global curvature). Note that this learning rate works in all smooth cases, regardless of the presence of strong convexity. We show that an analogous property (but not exactly descent) holds for LSUC functions by using 1/(local smoothness), preserving a similar rate. This is a remarkable property that does not hold in the non-smooth case, where one can show that the worst-case optimal stepsize is  $||x_0-x^{\ast}||_2/(G \sqrt{T})$ where $G$ is the Lipschitz constant. That, in the non-smooth case the optimal stepsize depends on unknown and global quantities ($||x_0-x^*||_2$), while in the smooth case we can just use a stepsize that depends on the local and observable behaviour of the function (local curvature). So, the smooth case is much easier. Now, our surrogate view essentially shows that we make any problem locally smooth, hence easier to be optimized.
> > >
> > > Regarding the previous statement re Polyak stepsize in relation to warm up and decay in ML problems. Are you asking if the Polyak stepsize and local curvature can explain why certain schedules are used in practice like warm up and decay? Given the widespread use of warm up and decaying learning rates it would be interesting to understand if there is a connection with the Polyak step. However, this would require assuming particular forms of the loss for example if one takes $f(x) = x^2/2$ then the Polyak step is a constant ½.  Additionally, the exact role of the warmup of the learning rates is currently unknown and probably only needed for specific optimizers, rather than being universal characteristics of the functions in deep learning and currently out scope for this paper.
> > > If we have misunderstood this part please let us know.

---

### Official Review · Reviewer_qnH4 · 2025-07-01

**Clarity:** 3
**Significance:** 3
**Originality:** 3
**Rating:** 4
**Confidence:** 4

**Summary:**

**Summary**

This paper establishes a new perspective on the Polyak stepsize in convex optimization. The idea is to consider a transformation of the original objective $\frac{1}{2}[f(x) - f(x^\star)]^2$, which inherits convexity from $f$ and naturally satisfies certain local curvature conditions. The authors demonstrate that gradient descent using the Polyak stepsize can be viewed as gradient descent on the transformed objective. The convergence analysis of the Polyak stepsize is established using this transformed objective in different settings, and some connections with previous literature on stochastic Polyak stepsize (SPS) are also discussed. Some negative results, including counterexamples, are presented to demonstrate the cost of not knowing the optimal value $f^\star$.

**Questions:**

**Questions**

1. Could you provide an example of a nonsmooth function that satisfies Definition 2?
2. On line 140, could you elaborate more on why $\phi(x_t) \rightarrow 0$ implies the last iterate converges?
3. On line 152, the quadratic growth condition misses a constant $\frac{1}{2}$; if you take $s = G = 1$ (e.g. $f(x) = |x|$), then the inequality becomes invalid.
4. In section 5, the function $h$ is defined as a general convex function. Is it necessary for $h$ to take the form of $(f(x) - a)_+$?  I did not find any restriction that $h$ should be relevant to $f$ at all, except that it is a nonnegative convex function. For example, taking $h(x) = 0$ would be a valid choice according to the discussion, but it does not yield a meaningful convergence guarantee.
5. It is claimed on line 225 that "[1] does not prove a guarantee for the nonsmooth setting". However, in Section 3.7 of [1], it is mentioned that results for the nonsmooth case are established in the appendix.
6. On line 269. Could you be more detailed about the mentioned example from [2]?

**Minor issues**

Please carefully proofread the paper.

1. Line 29

   adaptiveto => adaptive to

2. Line 87

   The definition of $L$-smoothness is not correct. Nonsmooth concave function also satisfies the bound.

3. Line 116

   Please be consistent with $f$ and $f(x)$ in the theorem statement.

4. Line 102, 120, 166

   Please be consistent with $g \in \partial f(x)$ or $g_x \in \partial f(x)$

5. Line 129

   "Let $\phi$ convex" is not clear; $\phi$ is convex by the composition rule of convex functions.

6. Line 131

   "summing this inequality over time" is not clear

7. Line 133

   $g_t$ and $g_{x_t}$ are inconsistent

8. Line 134

   "Now we use ... we obtain" is not clear

9. Line 161

   positive => nonnegative

10. Line 164

    The wording in definition 3 and 4 are inconsistent: e.g., say/will say; star-upper-curvature/star upper curvature

11. Line 173

    Missing $($

12. Line 175

    Why not directly use $x^\star = \arg \min \psi(u)$?

13. Algorithm 1, Line 3

    Tranform => Transform

14. Line 177, around Line 250

    $\lambda_t$ => $\lambda_{x_t}$ according to definition 4

15. Line 179

    tells => tell

16. Line 213

    $\xi$  and its bold face are inconsistent

17. Line 250, Line 305

    Paragraph titles are inconsistent

18. Line 263

    $T$ is used for both the operator and the total iteration count

19. Line 264

    The concept of clipping has not been introduced here

20. Line 243

    $x$ should be bold face

21. Line 291

    Inconsistent bold face issue again

22. Line 300

    The Lusin$(N^{-1})$ property only appears once in the paper without proper definition

23. Line 304

    under estimation => underestimation

24. Line 306

    $\min_x h(x, \xi) = 0$ is not clear

25. Line 763

    Bold face issue

26. Appendix

    $f^\star$ is inconsistent  with $f(x^\star)$ used in the main paper

27. Line 779

    functions => function

**References**

[1] Loizou, N., Vaswani, S., Laradji, I. H., & Lacoste-Julien, S. (2021, March). Stochastic polyak step-size for sgd: An adaptive learning rate for fast convergence. In *International Conference on Artificial Intelligence and Statistics* (pp. 1306-1314). PMLR.

[2] Bauschke, H. H., Wang, C., Wang, X., & Xu, J. (2018). Subgradient projectors: extensions, theory, and characterizations. *Set-Valued and Variational Analysis*, *26*, 1009-1078.

**Ethical Concerns:**

["NO or VERY MINOR ethics concerns only"]

**Final Justification:**

Overall, I raise my score to 4 given that the authors fix the typos/notation issues in the revision. But I'm still concerned that the paper fails to demonstrate new algorithmic improvements/variants. In particular, the counterexample does not rely on this new perspective, so the importance of the first part of the paper is not clear. I'm willing to raise my score further if the authors make (even preliminary) attempts in this direction, otherwise this paper remains borderline to me,

**Other minor issues**

1. Last iterate convergence

   This is more a consequence of Fejér monotonicity rather than just $\phi(x_t) \rightarrow 0$. Please clarify in the revision

2. In the revision, please add additional clarifications on 1) the requirement on the design of $h$ 2) the nonsmooth result compared to the SPS literature.

**Limitations:**

Yes.

**Paper Formatting Concerns:**

N.A.

**Quality:**

2

**Strengths And Weaknesses:**

**Strength**

This paper is overall well-written and provides an interesting new perspective on the Polyak stepsize. The results are supported by theoretical analysis. The motivation of the paper is clear, with theoretical results also connected to previous literature on SPS.

**Weaknesses**

Although I believe that the results in the paper provide new insights into the Polyak stepsize, I find the paper's presentation to be unsatisfactory:

1. Lack of practical algorithmic improvement

   Although the paper provides new insights into the Polyak stepsize and provides Algorithm 1 as a general template for new algorithmic variants, it lacks an in-depth discussion on whether these new variants are practically competitive. I recommend that the authors consider providing at least one instance of Algorithm 1 that has not appeared in the literature before and test it on toy examples. Otherwise, this perspective is of only theoretical interest.

2. There are a number of notation inconsistencies and typos in the paper. See minor issues and questions for more details.

Overall, I think the paper provides valuable insights into Polyak stepsize. However, given the issues above, I recommend borderline rejection, and I'm willing to raise my score if my concerns are addressed.

---

> ### Author Rebuttal · Authors · 2025-07-30
>
> We thank the Reviewer for carefully going over our paper. In particular, we will fix all the typos and minor mistakes in our notation.
> In the following we answer to the Reviewer’s questions
>
> 1. Lack of practical algorithmic improvement, this perspective is of only theoretical interest.
>
> Actually, writing a paper of “theoretical interest” was exactly our aim.
> Moreover, the call for papers of NeurIPS explicitly says
> “We also encourage in-depth analysis of existing methods that provide new insights in terms of their limitations or behaviour beyond the scope of the original work.”
> We agree that reviewers have freedom to interpret the reviewing guidelines, but we would still appreciate a more faithful interpretation.
>
> 1. Could you provide an example of a nonsmooth function that satisfies Definition 2?
>
> An easy example is in line 108: $f(x)=|x+2|+x^2/2$.
> We have that  $\partial f(x) = \\{1+x\\}$ for $x+2>0$, $[-3,-1]$ for $x=-2$, and $\\{-1+x\\}$ for $x+2<0$. Also $ x^{\ast}=-1 $ and $f(x^*)=1.5$.
> One can verify, for example by plotting the inequality, that this function is self-bounded for L=9 and not-smooth.
> In the same way, one can also show that f is L-LSUC with the better constant L =  2. Note, that one must also consider the worst case subgradient at $x=-2$.
>
> 2. On line 140, could you elaborate more on $\phi(\boldsymbol{x}_t)\to 0$ why implies the last iterate converges?
>
> Thank you for the question, the result follows from a classic argument with Fejér monotone sequences, we will add the details in the appendix. Briefly, from Lemma 1 we have that the distance to any solution is decreasing, that is, $\\{\boldsymbol{x}_t\\}_t$ is a Fejér monotone sequence with respect to the solution set. Since we have that $\phi(\boldsymbol{x}_t)\to 0$ and $\phi$ is continuous then we have that every limit  point $\boldsymbol{x}’$ is within the solution set since $\phi(\boldsymbol{x}')=0$. Therefore we can use the fact that if $\\{\boldsymbol{x}_t\\}_t$ is Fejér monotone with respect to the solution set and the set contains all the limit points of the sequence then the sequence must converge to a point in the solution set (see Theorem 8.16 in [Beck 17]).
>
> 3. On line 152, the quadratic growth condition misses a constant
>
> Correct, it is a typo.
>
> 4.  What are the conditions on h in Section 5?
>
> The observation that $h = 0$ “works” is correct. In this section we focus on bounds to the minimizer of the surrogate $\psi$ (line 167) which also coincides with the minimum of $h$ (since h is non-negative). To establish a rate on $f$, one needs to establish a connection between the minima of $h$ and $f$, which would not be possible if one were to take a constant function $h =0$. Because many variants are picking different h’s it makes sense to break up the analysis this way since, as we show, they are equivalent to subgradient descent on $\psi$ and so we should only expect these variants to converge to a minima of $\psi$ not $f$. Furthermore, having a separation reinforces the fact that there is a tradeoff between designing an “$h$” that is nice (e.g. $H(x^*) =0$) and one that connects to the original objective function $f$.
>
> 5. [1] has results for the nonsmooth case
>
> In line 225 we are referring to $SPS_{max}$, as mentioned in the following sentence [225-226], and implied by the context of our Theorem 3 (using stepsize from Algorithm 1).  In [1] there is no result for $SPS_{max}$ in the non-smooth case. In [1] they only have a nonsmooth result for SPS in the easier interpolation case. This is not a minor difference: They claim that “Using the proof techniques from the rest of the paper one can easily obtain convergence for the more general setting”, but this does not seem to be true. The technical reason is that the min function would introduce a non-convexity that would not allow the use of Jensen’s inequality. We show how to go around it, by lower bounding it with a convex function in line 805 in the supplemental material. So, a proof is indeed possible but not with the same tools they use. We will be more precise on this point and make it clear that we are referring to $SPS_{max}$.
>
> 6. Example from [2]
>
> The example discussed in [2] (Example  5.13) uses the functions $h(x) = e^{|x|}$, and $h(x)=e^{||x||^2/2}$. Both of which are lower bounded by 1. The technique used there is by calculating T explicitly and observing that it is not quasi firmly non-expansive.
>
>
> [1] Loizou, N., Vaswani, S., Laradji, I. H., & Lacoste-Julien, S. (2021, March). Stochastic polyak step-size for sgd: An adaptive learning rate for fast convergence. In International Conference on Artificial Intelligence and Statistics (pp. 1306-1314). PMLR.
>
> [2] Bauschke, H. H., Wang, C., Wang, X., & Xu, J. (2018). Subgradient projectors: extensions, theory, and characterizations. Set-Valued and Variational Analysis, 26, 1009-1078.
>
> [Beck 17] Beck, Amir. First-order methods in optimization. Society for Industrial and Applied Mathematics, 2017.

---

### Official Review · Reviewer_xpPF · 2025-07-03

**Clarity:** 3
**Significance:** 3
**Originality:** 4
**Rating:** 5
**Confidence:** 4

**Summary:**

This paper investigates the Polyak stepsize, a stepsize selection rule that has recently gained attention in the context of (stochastic) gradient descent. The authors point out that the gradient method with the Polyak stepsize can be interpreted as a simple gradient method applied to a surrogate function incorporating local curvature. Based on this perspective, they provide a unified analysis of (stochastic) gradient descent with the Polyak stepsize under various assumptions on the objective function. This analysis encompasses many existing results in the literature. In addition, the paper presents some negative results, such as the failure of the Polyak stepsize when the optimal value is inaccurately estimated.

**Questions:**

This is a question driven purely by personal curiosity: would it be possible to derive an algorithm that outperforms gradient descent with the Polyak stepsize by applying an **accelerated** gradient method to the surrogate function?

**Ethical Concerns:**

["NO or VERY MINOR ethics concerns only"]

**Final Justification:**

- While the proofs appear simple, the idea of introducing a surrogate function is likely nontrivial.
- It provides a unified analysis that encompasses many existing results.
- Although the paper offers a new interpretation that may serve as a foundation for future developments, it does not appear to provide techniques that are directly useful in practice.

**Limitations:**

Yes.

**Paper Formatting Concerns:**

None.

**Quality:**

3

**Strengths And Weaknesses:**

**Strengths**
- The paper addresses stepsize selection, a fundamental and important topic in this field.
- The perspective is novel and thought-provoking. While the proofs appear simple, the idea of introducing a surrogate function is likely nontrivial.
- It provides a unified analysis that encompasses many existing results.
- It clearly shows that the failure of the Polyak stepsize under violated assumptions stems from intrinsic limitations of the stepsize itself, rather than from shortcomings of the analytical technique.
- The manuscript is well-written and easy to follow.

**Weaknesses**
- Although the paper offers a new interpretation that may serve as a foundation for future developments, it does not appear to provide techniques that are directly useful in practice.

---

> ### Author Rebuttal · Authors · 2025-07-30
>
> We thank the reviewer for the feedback and for recognizing the value of our paper.
>
> - Would it be possible to derive an algorithm that outperforms gradient descent with the Polyak stepsize by applying an accelerated gradient method to the surrogate function?
>
> We are currently working on this question! Indeed, our surrogate view immediately suggests a way to make an accelerated version of Polyak stepsize, which would be difficult following the usual derivation. In fact, one can think of just using an accelerated scheme on the surrogate loss, using the LSUC constant instead of the global smoothness to set the stepsizes. That said, it is unlikely that we can get an accelerated result in all cases because that would imply a better convergence rate in the non-smooth case (see lines 145-146) that is impossible due to the known lower bound. Instead, we currently believe that one could obtain an algorithm whose rate is never worse than the Polyak one and it accelerates only when possible. We believe that our unifying view can open the door to many results of this form.

---

> > ### Comment · Reviewer_xpPF · 2025-08-06
> >
> > Thank you for your reply. That sounds very interesting. Follow-up work like that would further enhance the value of this research. I'm looking forward to your future work!

---

### Official Review · Reviewer_fAec · 2025-07-03

**Clarity:** 3
**Significance:** 3
**Originality:** 2
**Rating:** 5
**Confidence:** 3

**Summary:**

The authors study the Polyak method for optimizing a convex function f.  The authors note that from prior work, the Polyak method has qualitatively different convergence guarantees under various assumptions: for instance when f is convex/smooth convex/smooth strongly convex previous work showed 1/sqrt(T), 1/T, and linear convergence rates respectively.  These convergence guarantees look different, and use somewhat techniques.  The authors ask whether these convergence results can be unified under a single framework.

Suprisingly, they can be.  Specifically, they consider the "surrogate" function phi = (f - f^*)^2. When f is convex, they observe that phi has the $\|g_y\|^2$-LSUC property around $y$, so that the Polyak step size from y is just the inverse of the LSUC constant at y. Now using a simple convergence lemma that they prove for such step sizes (Lemma 1), they are able to deduce a variety of convergence results under various assumptions on f.

They also show how to extend their results to both the stochastic setting, and where only a lower bound of f^* is known.  Finally they prove a variety of lower bounds for Polyak methods, for example showing that the fixed points are unstable when the lower bound on f^* is not tight, and the step sizes are not appropriately clipped.

**Questions:**

1. It seems that the current approach is mainly using properties of phi, rather than f.  Is there hope of getting improved convergence result for Polyak in some situations where phi is convex, but f is not? (This might be a bit a niche, but I am not aware of such results.)

2. “Similarly, it is equally easy to obtain rates for Hölder-smooth functions. ”
I think it would be nice to state this explicitly in the main text.

Inequality at the end of section 4.  As a sanity check, the stated bound seems to imply that $(s^2/G^2 - 1/2)\|x_t - x^*\|^2 \leq \|x_{t+1} - x^*\|^2,$ which woiuld mean that $s^2/G^2 \leq1/2.$ I’m likely missing something, but I don’t see why this should be true without extra assumptions.  Can the authors clarify?

------- minor comments ------
- adaptiveto -- missing space
- Eq 1 -- this is the Polyak step rather than the stepsize
- "pionnering"
- Definition 1 -- should add a quantifier for x.  Is this a local condition, or for all x?
- Definition 2 -- check notation; x is doubly overloaded and g isn't defined.

**Ethical Concerns:**

["NO or VERY MINOR ethics concerns only"]

**Final Justification:**

The authors gave a reasonable response and my general assessment of the paper remains the same after reading the other reviews.  I think the observation that Polyak can be analyzed as gradient descent on a surrogate fuction is nice, and seems to unify various convergence results.  I think that this would be broadly interesting to an ML audience.

**Limitations:**

yes

**Quality:**

3

**Strengths And Weaknesses:**

Overall, I thought that this was a nice paper that sheds some interesting light on the Polyak method.  The main observation of the paper is that the Polyak method can be viewed as gradient descent on the surrogate function phi(x) = f(x)^2.  On the one hand, this is a fairly straightforward obsevation.  However it's memorable, and I think it should be more widely known, since it's a nice organizing principle for Polyak.

The main point of this observation is to streamline the convergnece anlayses for Polyak under a variety of assumptions of f.  The successfully achieve this by analyzing convergence of phi with step size given by 1/(LSUC constant).  This main result is given in their Lemma 1.  I suspect that a version of Lemma 1 could be rewritten directly in terms of f, and that this would give a similarly general result that could be applied for various f to obtain the main results.  So I'm not completely clear on whether considering the surrogate is entirely crucial here.  With that said, I think Lemma 1 is more naturally applied to phi, so I like the results as stated.  Working with phi is just perhaps more of an organizing principle than something that will lead to new convergence results that couldn't easily be obtained otherwise.

I thought the negative results were nice.  These really seem to be an improvement over existing work.

---

> ### Author Rebuttal · Authors · 2025-07-30
>
> We thank the reviewer for the feedback and for recognizing the fact that our unifying surrogate view is “memorable” and “it should be more widely known”.
>
> - Is there hope of getting improved convergence results for Polyak in some situations where $\phi$ is convex, but $f$ is not?
>
> Yes! In fact, we wrote about it in Remark 2: We consider a stochastic optimization problem where one of the functions is non-convex, yet the surrogate of the function is convex. As far as we know, this is the first guarantee of this kind for Polyak stepsizes.
>
> - Rates for Hölder-smooth functions
>
> We can add them, they are an immediate corollary of Theorem 7 in the Supplemental Material.
>
> - Inequality at the end of section 4
>
> It’s a typo, thanks for pointing it out. In particular, quadratic growth implies the presence of a factor 1/2 on the first two terms of the chain of inequalities. With that term, we only need $s^2/G^2\leq 1$, that is true.

---

> > ### Comment · Reviewer_fAec · 2025-08-09
> >
> > Thank you for the clarifications!  "Rates for Hölder-smooth functions. We can add them, they are an immediate corollary of Theorem 7 in the Supplemental Material."  I think this is optional, so the authors should do what they feel is best for the presentation.

---

### Comment · Area_Chair_dx1V · 2025-08-04

Dear reviewers,

Thank you for your valuable time and your expertise in reviewing. Engaging with authors is really important, and allows both authors and reviewers to gain deeper understanding of cutting edge topics. This is a unique opportunity of interaction in our community.

The author rebuttal phase is about to close, and we kindly request your prompt attention to ensure a thorough discussion.

The discussion period **ends in less than 3 days** (on Aug. 6, 11:59pm AOE ). To maintain the review timeline, we ask that you:

- Review the rebuttals,

- Engage in any ongoing discussion with fellow reviewers/authors (if applicable),

- Finalize your assessment.

If you have already completed this step, we sincerely appreciate your efforts.

Thank you for your collaboration!

Best regards,

AC

---

### Note · Authors · 2025-08-12

We would like to thank all reviewers for their detailed comments and suggestions that will improve the paper.

We are also happy that all reviewers found valuable novel insight on the Polyak stepsize from our new surrogate perspective. In our work we have set out to better understand the behaviour of different Polyak variants: why they work and why they fail. To this end, we show that Polyak’s stepsize and its variants can be viewed as gradient descent on a surrogate loss with a stepsize that is 1/local-curvature of the surrogate.

This perspective has allowed us to:

+ Provide a unified simple theory where descent using local curvature automatically gives rates in many settings: deterministic, stochastic, non-smooth, smooth, sharp minima, strongly convex, Holder-smoothness.
+ Show a new connection between neighbourhoods of convergence and approximate local curvature:  A neighbourhood of convergence is due to the inexact local smoothness of the surrogate.
+ Prove new negative results demonstrating that neighbourhoods of convergence are unavoidable in general if the minimum of the surrogate is not zero. These results show:
   - Interpolation is not the only reason for neighbourhoods of convergence, such results can occur even in the deterministic setting.
   - Fundamental issues arise such as instability of minima.
   - Unexpected behaviour can happen such as cycles.
   - An intuitive connection with local curvature of the surrogate, these issues occur when the stepsize on the surrogate can get arbitrarily large with respect to local curvature.

We believe that our results and perspective brings value to the community by providing a better understanding of the Polyak stepsize and its variants. Moreover, we believe these insights will be invaluable for pursuing new directions of research that can better leverage local smoothness of the surrogate, such as, for example, an accelerated version of Polyak’s stepsize.

---

### Decision · Program_Chairs · 2025-09-17

**Decision:**

Accept (poster)

**Comment:**

The paper at hand proposes novel theoretical perspectives on (stochastic) Polyak stepsizes. The paper is pleasant to read, and the authors take us step-by-step into an interesting correspondence between PS and direct optimization of a surrogate function with the same minimizer, but interesting curvature properties to which PS directly adapts.

All reviewers agree that the paper is correct and interesting for the optimization community, and hence I recommend accepting this for the conference.

However, I believe the authors should take the following points into account for the camera-ready:

1) As suggested by Reviewer AKhF, I would strongly recommend placing illustrations and toy simulations in the paper to guide the reader through the text and the derivations. Especially, this would enhance the intuition around the curvature adaptations claims. I also agree with Reviewer AKhF that exploring new surrogate functions and understanding the effects of choosing such functions would be highly desirable. The authors may be able to expand a bit on this in their revision.

2) I would include a few more citations linking the Polyak stepsize to surrogate functions (e.g., squaring the loss), in particular [A,B] below.

3) Improve clarity: I think it might be helpful to actually structure a bit more the contribution section, to better guide the reader in the exploration of your work.

[A] Chen, Pei. "Hessian matrix vs. Gauss–Newton hessian matrix." SIAM Journal on Numerical Analysis 49.4 (2011): 1417-1435.
[B] Orvieto, Antonio, and Lin Xiao. "An adaptive stochastic gradient method with non-negative Gauss-Newton stepsizes." arXiv preprint arXiv:2407.04358 (2024).